# Statistical and Computational Complexities of BFGS Quasi-Newton Method for Generalized Linear Models

**Qiujiang Jin**  *qiujiang@austin.utexas.edu*
*Department of Electrical and Computer Engineering*
*University of Texas at Austin*

**Tongzheng Ren**  *tongzheng@utexas.edu*
*Department of Computer Science*
*University of Texas at Austin*

**Nhat Ho**  *minhnhat@utexas.edu*
*Department of Statistics and Data Sciences*
*University of Texas at Austin*

**Aryan Mokhtari**  *mokhtari@austin.utexas.edu*
*Department of Electrical and Computer Engineering*
*University of Texas at Austin*

**Reviewed on OpenReview:** *https://openreview.net/forum?id=PIL3YWXmx2*

## Abstract

The gradient descent (GD) method has been used widely to solve parameter estimation in generalized linear models (GLMs), a generalization of linear models when the link function can be non-linear. In GLMs with a polynomial link function, it has been shown that in the high signal-to-noise ratio (SNR) regime, due to the problem's strong convexity and smoothness, GD converges linearly and reaches the final desired accuracy in a logarithmic number of iterations. In contrast, in the low SNR setting, where the problem becomes locally convex, GD converges at a slower rate and requires a polynomial number of iterations to reach the desired accuracy. Even though Newton's method can be used to resolve the flat curvature of the loss functions in the low SNR case, its computational cost is prohibitive in high-dimensional settings as it is $\mathcal{O}(d^3)$, where $d$ the is the problem dimension. To address the shortcomings of GD and Newton's method, we propose the use of the BFGS quasi-Newton method to solve parameter estimation of the GLMs, which has a per iteration cost of $\mathcal{O}(d^2)$. When the SNR is low, for GLMs with a polynomial link function of degree $p$, we demonstrate that the iterates of BFGS converge linearly to the optimal solution of the population least-square loss function, and the contraction coefficient of the BFGS algorithm is comparable to that of Newton's method. Moreover, the contraction factor of the linear rate is independent of problem parameters and only depends on the degree of the link function $p$. Also, for the empirical loss with $n$ samples, we prove that in the low SNR setting of GLMs with a polynomial link function of degree $p$, the iterates of BFGS reach a final statistical radius of $\mathcal{O}((d/n)^{\frac{1}{2p+2}})$ after at most $\log(n/d)$ iterations. This complexity is significantly less than the number required for GD, which scales polynomially with $(n/d)$.

## 1 Introduction

In supervised machine learning, we are given a set of $n$ independent samples denoted by $X_1, \ldots, X_n$ with corresponding labels $Y_1, \ldots, Y_n$, that are drawn from some unknown distribution and our goal is to train a model that maps the feature vectors to their corresponding labels. We assume that the data is generated

according to distribution $\mathcal{P}_{\theta^*}$ which is parameterized by a ground truth parameter $\theta^*$. Our goal as the learner is to find $\theta^*$ by solving the empirical risk minimization (ERM) problem defined as

$$\min_{\theta \in \mathbb{R}^d} \mathcal{L}_n(\theta) := \frac{1}{n} \sum_{i=1}^{n} \ell(\theta; (X_i, Y_i)), \tag{1}$$

where $\ell(\theta; (X_i, Y_i))$ is the loss function that measures the error between the predicted output of $X_i$ using parameter $\theta$ and its true label $Y_i$. If we define $\theta_n^*$ as an optimal solution of the above optimization problem, i.e., $\theta_n^* \in \arg\min_{\theta \in \mathbb{R}^d} \mathcal{L}_n(\theta)$, it can be considered as an approximation of the ground-truth solution $\theta^*$, where $\theta^*$ is also a minimizer of the population loss defined as

$$\min_{\theta \in \mathbb{R}^d} \mathcal{L}(\theta) := \mathbb{E}\left[\ell(\theta; (X, Y))\right]. \tag{2}$$

If one can solve the empirical risk efficiently, the output model could be close to $\theta^*$, when $n$ is sufficiently large. Several works have studied the complexity of iterative methods for solving ERM or directly the population loss, for the case that the objective function is convex or strongly convex with respect to $\theta$ (Balakrishnan et al., 2017; Ho et al., 2020; Loh & Wainwright, 2015; Agarwal et al., 2012; Yuan & Zhang, 2013; Dwivedi et al., 2020b; Hardt et al., 2016; Candes et al., 2011). However, when we move beyond linear models, the underlying loss becomes non-convex and therefore the behavior of iterative methods could substantially change, and it is not even clear if they can reach a neighborhood of a global minimizer of the ERM problem.

The focus of this paper is on the generalized linear model (GLM) (Carroll et al., 1997; Netrapalli et al., 2015; Fienup, 1982; Shechtman et al., 2015; Feiyan Tian, 2021) where the labels and features are generated according to a polynomial link function and we have $Y_i = (X_i^\top \theta^*)^p + \zeta_i$, where $\zeta_i$ is an additive noise and $p \geq 2$ is an integer. Due to nonlinear structure of the generative model, even if we select a convex loss function $\ell$, the ERM problem denoted to the considered GLM could be non-convex with respect to $\theta$. Interestingly, depending on the norm of $\theta^*$, the curvature of the ERM problem and its corresponding population risk minimization problem could change substantially. More precisely, in the case that $\|\theta^*\|$ is sufficiently large, which we refer to this case as the high signal-to-noise ratio (SNR) regime, the underlying population loss of the problem of interest is locally strongly convex and smooth. On the other hand, in the regime that $\|\theta^*\|$ is close to zero, denoted by the low SNR regime, the underlying problem is neither strongly convex nor smooth, and in fact, it is ill-conditioned.

These observations lead to the conclusion that in the high SNR setting, due to strong convexity and smoothness of the underlying problem, gradient descent (GD) reaches the desired accuracy exponentially fast and overall it only requires logarithmic number of iterations. However, in the low SNR case, as the problem becomes locally convex, GD converges at a sublinear rate and thus requires polynomial number of iterations in terms of the sample size.

To address this issue, Ren et al. (2022a) advocated the use of GD with the Polyak step size to improve GD's convergence in low SNR scenarios. hey demonstrated that the number of iterations could become logarithmic with respect to the sample size, when GD is deployed with the Polyak step size. However, such a method still remains a first-order algorithm and lacks any curvature approximation. Consequently, its overall complexity is directly proportional to the condition number of the problem. This, in turn, depends on both the condition number of the feature vectors' covariance and the norm $|\theta^*|$. As a result, in low SNR settings, the problem becomes ill-conditioned with a large condition number, and hence GD with the Polyak step size could be very slow. Furthermore, the implementation of the Polyak step size necessitates access to the optimal value of the objective function. As precise estimation of this optimal objective function value may not always be feasible, any inaccuracies could potentially lead to a reduced convergence rate for GD employing the Polyak step size.

Another alternative is to use a different distance metric instead of an adaptive step size to improve the convergence of GD for the low SNR setting. More precisely, Lu et al. (2018) have shown that the mirror descent method with a proper distance-generating function can solve the population loss corresponding to the low SNR setting at a linear rate. However, the linear convergence rate of mirror descent, similar to GD with Polyak step size, also depends on the condition number of the problem, and hence could lead to a slow convergence rate.

A natural approach to handle the ill-conditioning issue in the low SNR case as well as eliminating the need to estimate the optimal function value is the use of Newton's method. As we show in this paper, this idea indeed addresses the issue of poor curvature of the problem and leads to an exponentially fast rate with contraction factor $\frac{2p-2}{2p-1}$ in the population case, where $p$ is the degree of the polynomial link function. Moreover, in the high SNR setting, Newton's method converges at a quadratic rate as the problem is strongly convex and smooth. Alas, these improvements come at the expense of increasing the computational complexity of each iteration to $\mathcal{O}(d^3)$ which is indeed more than the per iteration computational cost of GD that is $\mathcal{O}(d)$. These points lead to this question:

> *Is there a computationally-efficient method that performs well in both high and low SNR settings at a reasonable per iteration computational cost?*

**Contributions.** In this paper, we address this question and show that the BFGS method is capable of achieving these goals. BFGS is a quasi-Newton method that approximates the objective function curvature using gradient information and its per iteration cost is $\mathcal{O}(d^2)$. It is well-known that it enjoys a superlinear convergence rate that is independent of condition number in strongly convex and smooth settings, and hence, in the high SNR setting it outperforms GD. In the low SNR setting, where the Hessian at the optimal solution could be singular, we show that the BFGS method converges linearly and outperforms the sublinear convergence rate of GD. Next, we formally summarize our contributions.

- **Infinite sample, low SNR:** For the infinite sample case where we minimize the population loss, we show that in the low SNR case the iterates of BFGS converge to the ground truth $\theta^*$ at an exponentially fast rate that is independent of all problem parameters except the power of link function $p$. We further show that the linear convergence contraction coefficient of BFGS is comparable to that of Newton's method. This convergence result of BFGS is also of general interest as it provides the first global linear convergence of BFGS without line-search for a setting that is neither strictly nor strongly convex.

- **Finite sample, low SNR:** By leveraging the results developed for the population loss of the low SNR regime, we show that in the finite sample case, the BFGS iterates converge to the final statistical radius $\mathcal{O}(1/n^{1/(2p+2)})$ within the true parameter after a logarithmic number of iterations $\mathcal{O}(\log(n))$. It is substantially lower than the required number of iterations for fixed-step size GD, which is $\mathcal{O}(n^{(p-1)/p})$, to reach a similar statistical radius. This improvement is the direct outcome of the linear convergence of BFGS versus the sublinear convergence rate of GD in the low SNR case. Further, while the iteration complexity of BFGS is comparable to the logarithmic number of iterations of GD with Polyak step size, we show that BFGS removes the dependency of the overall complexity on the condition number of the problem as well as the need to estimate the optimal function value.

- **Experiments:** We conduct numerical experiments for both infinite and finite sample cases to compare the performance of GD (with constant step size and Polyak step size), Newton's method and BFGS. The provided empirical results are consistent with our theoretical findings and show the advantages of BFGS in the low SNR regime.

**Outline.** In Section 2, we discuss the BFGS quasi-Newton method. Section 3 details three scenarios in Generalized Linear Models (GLMs): low, middle, and high SNR regimes, outlining the characteristics of the population loss in each. Section 4 explores BFGS's convergence in low SNR settings, highlighting its linear convergence rate, a marked improvement over gradient descent's sublinear rate. This section also compares the convergence rates of BFGS and Newton's method. Section 5 applies these insights to establish the convergence results of BFGS for the empirical loss $\mathcal{L}_n$ in the low SNR regime. Lastly, our numerical experiments are presented in Section 6.

## 2 BFGS algorithm

In this section, we review the basics of the BFGS quasi-Newton method, which is the main algorithm we analyze. Consider the case that we aim to minimize a differentiable convex function $f : \mathbb{R}^d \to \mathbb{R}$. The BFGS

update is given by

$$\theta_{k+1} = \theta_k - \eta_k H_k \nabla f(\theta_k), \qquad \forall k \geq 0, \tag{3}$$

where $\eta_k$ is the step size and $H_k \in \mathbb{R}^{d \times d}$ is a positive definite matrix that aims to approximate the true Hessian inverse $\nabla^2 f(\theta_k)^{-1}$. There are several approaches for approximating $H_k$ leading to different quasi-Newton methods, (Conn et al., 1991; Broyden, 1965; Broyden et al., 1973; Gay, 1979; Davidon, 1959; Fletcher & Powell, 1963; Broyden, 1970; Fletcher, 1970; Goldfarb, 1970; Shanno, 1970; Nocedal, 1980; Liu & Nocedal, 1989), but in this paper, we focus on the celebrated BFGS method, in which $H_k$ is updated as

$$H_k = \left( I - \frac{s_{k-1} u_{k-1}^\top}{s_{k-1}^\top u_{k-1}} \right) H_{k-1} \left( I - \frac{u_{k-1} s_{k-1}^\top}{s_{k-1}^\top u_{k-1}} \right) + \frac{s_{k-1} s_{k-1}^\top}{s_{k-1}^\top u_{k-1}}, \qquad \forall k \geq 1, \tag{4}$$

where the variable variation $s_k$ and gradient displacement $u_k$ are defined as

$$s_{k-1} := \theta_k - \theta_{k-1}, \qquad u_{k-1} := \nabla f(\theta_k) - \nabla f(\theta_{k-1}), \tag{5}$$

for any $k \geq 1$ respectively. The logic behind the update in (4) is to ensure that the Hessian inverse approximation $H_k$ satisfies the secant condition $H_k u_{k-1} = s_{k-1}$, while it stays close to the previous approximation matrix $H_{k-1}$. The update in (4) only requires matrix-vector multiplications, and hence, the computational cost per iteration of BFGS is $\mathcal{O}(d^2)$.

The main advantage of BFGS is its fast superlinear convergence rate under the assumption of strict convexity,

$$\lim_{k \to \infty} \frac{\|\theta_k - \theta_{opt}\|}{\|\theta_{k-1} - \theta_{opt}\|} = 0,$$

where $\theta_{opt}$ is the optimal solution. Previous results on the superlinear convergence of quasi-Newton methods were all asymptotic, until the recent advancements on the non-asymptotic analysis of quasi-Newton methods (Rodomanov & Nesterov, 2021a;b;c; Jin & Mokhtari, 2020; Jin et al., 2022; Ye et al., 2021; Lin et al., 2021a;b). For instance, Jin & Mokhtari (2020) established a local superlinear convergence rate of $(1/\sqrt{k})^k$ for BFGS. However, all these superlinear convergence analyses require the objective function to be smooth and strictly or strongly convex. Alas, these conditions are not satisfied in the low SNR setting, since the Hessian at the optimal solution could be singular, and hence the loss function is neither strongly convex nor strictly convex; we further discuss this point in Section 3. This observation implies that we need novel convergence analyses to study the behavior of BFGS in the low SNR setting, as we discuss in the upcoming sections.

## 3 Generalized linear model with polynomial link function

In this section, we formally present the generalized linear model (GLM) setting that we consider in our paper, and discuss the low and high SNR settings and optimization challenges corresponding to these cases. Consider the case that the feature vectors are denoted by $X \in \mathbb{R}^d$ and their corresponding labels are denoted by $Y \in \mathbb{R}$. Suppose that we have access to $n$ sample points $(Y_1, X_1), (Y_2, X_2), \ldots, (Y_n, X_n)$ that are i.i.d. samples from the following generalized linear model with polynomial link function of power $p$ (Carroll et al., 1997), i.e.,

$$Y_i = (X_i^\top \theta^*)^p + \zeta_i, \tag{6}$$

where $\theta^*$ is a true but unknown parameter, $p \in \mathbb{N}$ is a given power, and $\zeta_1, \ldots, \zeta_n$ are independent Gaussian noises with zero mean and variance $\sigma^2$. The Gaussian assumption on the noise is for the simplicity of the discussion and similar results hold for the sub-Gaussian i.i.d. noise case. Furthermore, we assume the feature vectors are generated as $X \sim \mathcal{N}(0, \Sigma)$ where $\Sigma \in \mathbb{R}^{d \times d}$ is a symmetric positive definite matrix. Here we focus on the settings that $p \in \mathbb{N}^+$ and $p \geq 2$.

The above class of GLMs with polynomial link functions arise in several settings. When $p = 1$, the model in (6) is the standard linear regression model, and for the case that $p = 2$, the above setup corresponds to the phase retrieval model (Fienup, 1982; Shechtman et al., 2015; Candes et al., 2011; Netrapalli et al., 2015), which has found applications in optical imaging, x-ray tomography, and audio signal processing. Moreover, the analysis of GLMs with $p \geq 2$ also serves as the basis of the analysis on other popular statistical models.

For example, as shown by Yi & Caramanis (2015); Wang et al. (2015); Xu et al. (2016); Balakrishnan et al. (2017); Daskalakis et al. (2017); Dwivedi et al. (2020a); Kwon et al. (2019); Dwivedi et al. (2020b); Kwon et al. (2021), the local landscape of log-likelihood for Gaussian mixture models and mixture linear regression models are identical to GLMs for $p = 2$.

In the case that the polynomial link function parameter in the GLM model in (6) is $p = 2$, by adapting similar arguments from Kwon et al. (2021) under the symmetric two-component location Gaussian mixture, there are essentially three regimes for estimating the true parameter $\theta^*$: Low signal-to-noise ratio (SNR) regime: $\|\theta^*\|/\sigma \leq C_1(d/n)^{1/4}$ where $d$ is the dimension, $n$ is the sample size, and $C_1$ is a universal constant; Middle SNR regime: $C_1(d/n)^{1/4} \leq \|\theta^*\|/\sigma \leq C$ where $C$ is a universal constant; High SNR regime: $\|\theta^*\|/\sigma \geq C$. The main idea is that with different $\theta^*$, the optimization landscape of the parameter estimation problem changes. By generalizing the insights from the case $p = 2$, we define the following regimes for any $p \geq 2$:

- (i) Low SNR regime: $\|\theta^*\|/\sigma \leq C_1(d/n)^{1/(2p)}$ where $d$ is the dimension, $n$ is the sample size, and $C_1$ is a universal constant;

- (ii) Middle SNR regime: $C_1(d/n)^{1/(2p)} \leq \|\theta^*\|/\sigma \leq C$ where $C$ is a universal constant;

- (iii) High SNR regime: $\|\theta^*\|/\sigma \geq C$.

Note that, the rate $(d/n)^{1/2p}$ that we use to define the SNR regimes is from the statistical rate of estimating the true parameter $\theta^*$ when $\theta^*$ approaches 0. Next, we provide insight into the landscape of the least-square loss function for each regime. In particular, given the GLM in (6), the sample least-square takes the following form:

$$\min_{\theta \in \mathbb{R}^d} \mathcal{L}_n(\theta) := \frac{1}{n} \sum_{i=1}^n \left( Y_i - \left( X_i^\top \theta \right)^p \right)^2. \tag{7}$$

To obtain insight about the landscape of the empirical loss function $\mathcal{L}_n$, a useful approach is to consider that function by its population version, which we refer to as population least-square loss function:

$$\min_{\theta \in \mathbb{R}^d} \mathcal{L}(\theta) := \mathbb{E}[\mathcal{L}_n(\theta)], \tag{8}$$

where the outer expectation is taken with respect to the data.

**High SNR regime.** In the setting that the ground truth parameter has a relatively large norm, i.e., $\|\theta^*\| \geq C$ for some constant $C > 0$ that only depends on $\sigma$, the population loss in (8) is locally strongly convex and smooth around $\theta^*$. More precisely, when $\|\theta - \theta^*\|$ is small, using Taylor's theorem we have

$$(X^\top \theta)^p - (X^\top \theta^*)^p = p(X^\top \theta^*)^{p-1} X^\top (\theta - \theta^*) + o(\|\theta - \theta^*\|).$$

Hence, in a neighborhood of the optimal solution, the objective in (8) can be approximated as

$$\begin{aligned}
\mathcal{L}(\theta) &= \mathbb{E}[(Y - (X^\top \theta)^p)^2] = \mathbb{E}[((X^\top \theta^*)^p + \zeta - (X^\top \theta)^p)^2] \\
&= \mathbb{E}[((X^\top \theta^*)^p - (X^\top \theta)^p)^2] + \mathbb{E}[2\zeta((X^\top \theta^*)^p - (X^\top \theta)^p)^2] + \mathbb{E}[\zeta^2] = \mathbb{E}[((X^\top \theta^*)^p - (X^\top \theta)^p)^2] + \sigma^2 \\
&= p^2(\theta - \theta^*)^\top \mathbb{E}_X \left[ X(X^\top \theta^*)^{2p-2} X^\top \right] (\theta - \theta^*) + \sigma^2 + o(\|\theta - \theta^*\|^2).
\end{aligned}$$

Indeed, if $\|\theta^*\| \geq C\sigma$ the above function behaves as a quadratic function that is smooth and strongly convex, assuming that $o(\|\theta - \theta^*\|^2)$ is negligible. As a result, the iterates of gradient descent (GD) converge to the solution at a linear rate and it requires $\kappa \log(1/\epsilon)$ to reach an $\epsilon$-accurate solution, where $\kappa$ depends on the conditioning of the covariance matrix $\Sigma$ and the norm of $\theta^*$. In this case, BFGS converges superlinearly to $\theta^*$ and the rate would be independent of $\kappa$, while the cost per iteration would be $\mathcal{O}(d^2)$.

**Low SNR regime.** As mentioned above, in the high SNR case, GD has a fast linear rate. However, in the low SNR case where $\|\theta^*\|$ is small and $\|\theta^*\| \leq C_1(d/n)^{1/(2p)}$, the strong convexity parameter approaches zero when the sample size $n$ goes to infinity and the problem becomes ill-conditioned. In this case, we deal with a *function that is only convex and its gradient is not Lipschitz continuous*. To better elaborate on this point, let us focus on the case that $\theta^* = 0$ as a special case of the low SNR setting. Considering the underlying distribution of $X$, which is $X \sim \mathcal{N}(0, \Sigma)$, for such a low SNR case, the population loss can be written as

$$\mathcal{L}(\theta) = \mathbb{E}_X \left[ (X^\top \theta)^{2p} \right] + \sigma^2 = (2p-1)!! \|\Sigma^{1/2} \theta\|^{2p} + \sigma^2. \tag{9}$$

Since we focus on $p \geq 2$ it can be verified that $\mathcal{L}(\theta)$ is not strongly convex in a neighborhood of the solution $\theta^* = 0$. For this class of functions, it is well-known that GD with constant step size would converge at a sublinear rate, and hence GD iterates require polynomial number of iterations to reach the final statistical radius. In the next section, we study the behavior of BFGS for solving the low SNR setting and showcase its advantage compared to GD.

**Middle SNR regime.** Different from the low and high SNR regimes, the middle SNR regime is generally harder to analyze as the landscapes of both the population and sample least-square loss functions are complex. Adapting the insight from middle SNR regime of the symmetric two-component location Gaussian mixtures from Kwon et al. (2021), for the middle SNR setting of the generalized linear model, the eigenvalues of the Hessian matrix of the population least-square loss function approach 0 and their vanishing rates depend on some increasing function of $\|\theta^*\|$. The optimal statistical rate of the optimization algorithms, such as gradient descent algorithm, for solving $\theta^*$ depends strictly on the tightness of these vanishing rates in terms of $\|\theta^*\|$, which are non-trivial to obtain. In fact, to the best of our knowledge, there is no result on the convergence of iterative methods (such as GD or its variants) for GLMs with a polynomial link function in the middle SNR. Hence, we leave the study of BFGS for this setting as a future work.

## 4 Convergence analysis in the low SNR regime: Population loss

In this section, we focus on the convergence properties of BFGS for the population loss in the low SNR case introduced in (9). This analysis provides an intuition for the analysis of the finite sample case that we discuss in Section 5, as we expect these two loss functions to be close to each other when the number of samples $n$ is sufficiently large. In this section, instead of focusing on the population loss within the low SNR regime, as described in (9), we shift our attention to a more encompassing objective function, $f$. Detailed in the following expression, this function encompasses the population loss function $\mathcal{L}$ with $\theta^* = 0$ as a specific example. This approach enables us to present our results in the most general setting possible. Specifically, we examine the function $f : \mathbb{R}^d \to \mathbb{R}$, defined as follows:

$$\min_{\theta \in \mathbb{R}^d} f(\theta) = \|A\theta - b\|^q, \tag{10}$$

where $A \in \mathbb{R}^{m \times d}$ is a matrix, $b \in \mathbb{R}^m$ is a given vector, and $q \in \mathbb{Z}$ satisfies $q \geq 4$. We should note that for $q \geq 4$, the considered objective is not strictly convex because the Hessian is singular when $A\theta = b$. Indeed, if we set $m = d$ and further let $A$ be $\Sigma^{1/2}$ and choose $b = A\theta^* = 0$, then we recover the problem in (9) for $q = 2p$. Note that since we focus on the generalized linear model with the polynomial link function, which necessitates that $p$ be an integer, the parameter $q = 2p$ is also an integer.

Notice that the problem in (10), which serves as a surrogate for the finite sample problem that we plan to study in the next section, has the same solution set as the quadratic problem of minimizing $\|A\theta - b\|^2$ with solution $(A^\top A)^{-1} A^\top b$ when $A^\top A$ is invertible. Given this point, one might suggest that instead of minimizing (10) we could directly solve the quadratic problem which is indeed much easier to solve. This point is valid, but the goal of this section is not to efficiently solve problem (10) itself. Our goal is to understand the convergence properties of the BFGS method for solving the problem in (10) with the hope that it will provide some intuitions for the convergence analysis of BFGS for the empirical loss (7) in the low SNR regime. As we will discuss in Section 5, the convergence analysis of BFGS on the population loss which is a special case of (10) is closely related to the one for the empirical loss in (7).

**Remark 1.** *One remark is that population loss in (9) holds for the restrictive assumption of $\theta^* = 0$, which is only a special case of the general low SNR regime of $\|\theta^*\| \leq C_1(d/n)^{1/(2p)}$. Our ultimate goal is to analyze the convergence behavior of the BFGS method applied to the empirical loss (7) of GLM problems in the low SNR regime. The errors between gradients and Hessians of the population loss with $\theta^* = 0$ and $\|\theta^*\| \leq C_1(d/n)^{1/(2p)}$ are upper bounded by the corresponding statistical errors between the population loss (8) and the empirical loss (7) in the low SNR regime, respectively. Therefore, the errors between iterations of applying BFGS to the population loss with $\theta^* = 0$ and $\|\theta^*\| \leq C_1(d/n)^{1/(2p)}$ are negligible compared to the statistical errors. Instead of directly analyzing BFGS for the population loss (8) in the general low SNR regime, studying the convergence properties of BFGS for the population loss (9) with $\theta^* = 0$ can equivalently lay foundations for*

*the convergence analysis of BFGS for the empirical loss* (7) *in the low SNR regime, which is the main target of this paper. The details can be found in Appendix A.4.*

Our results in this section are of general interest from an optimization perspective, since there is no global convergence theory (without line-search) for the BFGS method in the literature when the objective function is not strictly convex. Our analysis provides one of the first results for such general settings. That said, it should be highlighted that our results do not hold for general convex problems, and we do utilize the specific structure of the considered convex problem to establish our results. The following examples illustrate some of the specific structures that we exploit to establish our result.

**Assumption 1.** *There exists $\hat{\theta} \in \mathbb{R}^d$, such that $b = A\hat{\theta}$. In other words, $b$ is in the range of matrix $A$.*

This assumption implies that the problem in (10) is realizable, $\hat{\theta}$ is an optimal solution, and the optimal function value is zero. This assumption is satisfied in our considered low SNR setting in (9) as $\theta^* = 0$ which implies $b = 0$.

**Assumption 2.** *The matrix $A^\top A \in \mathbb{R}^{d \times d}$ is invertible. This is equivalent to $A^\top A \succ 0$.*

The above assumption is also satisfied for our considered setting as we assume that the covariance matrix for our input features is positive definite. Combining Assumptions 1 and 2, we conclude that $\hat{\theta}$ is the unique optimal solution of problem (10). Next, we state the convergence rate of BFGS for solving problem (10) under the disclosed assumptions. The proof of the next theorem is available in Appendix A.1.

**Theorem 1.** *Consider the BFGS method in* (3)-(5). *Suppose Assumptions 1 and 2 hold, and the initial Hessian inverse approximation matrix is selected as $H_0 = \nabla^2 f(\theta_0)^{-1}$, where $\theta_0 \in \mathbb{R}^d$ is an arbitrary initial point. If the step size is $\eta_k = 1$ for all $k \geq 0$, then the iterates of BFGS converge to the optimal solution $\hat{\theta}$ at a linear rate of*

$$\|\theta_k - \hat{\theta}\| \leq r_{k-1}\|\theta_{k-1} - \hat{\theta}\|, \quad \forall k \geq 1, \tag{11}$$

*where the contraction factors $r_k \in [0, 1)$ satisfy*

$$r_0 = \frac{q-2}{q-1}, \qquad r_k = \frac{1 - r_{k-1}^{q-2}}{1 - r_{k-1}^{q-1}}, \quad \forall k \geq 1. \tag{12}$$

Theorem 1 shows that the iterates of BFGS converge globally at a linear rate to the optimal solution of (10). This result is of interest as it illustrates the iterates generated by BFGS converge globally without any line search scheme and the step size is fixed as $\eta_k = 1$ for any $k \geq 0$. Moreover, the initial point $\theta_0$ is arbitrary and there is no restriction on the distance between $\theta_0$ and the optimal solution $\hat{\theta}$.

We should highlight the above linear convergence result and our convergence analysis both rely heavily on the distinct structure of problem (10) and may not hold for a general convex minimization problem. Specifically, it can be shown that if we had access to the exact Hessian and could perform a Newton's update to solve the problem in (10), then the error vectors $(\theta_k - \hat{\theta})_{k \geq 0}$ would all be parallel. This property arises due to the fact that for the problem in (10) the Newton direction is $\nabla^2 f(\theta_{k-1})^{-1}\nabla f(\theta_{k-1}) = (\theta_{k-1} - \hat{\theta}) - \frac{q-2}{q-1}(\theta_{k-1} - \hat{\theta})$. Consequently, the next error vector $\theta_{k-1} - \hat{\theta}$ computed by running one step of Newton would satisfy $\theta_k - \hat{\theta} = \frac{q-2}{q-1}(\theta_{k-1} - \hat{\theta})$. Therefore, the error vector at time $k$, denoted by $\theta_k - \hat{\theta}$, is parallel to the previous error vector $\theta_{k-1} - \hat{\theta}$, with the only difference being that its norm is reduced by a factor of $\frac{q-2}{q-1}$. Using an induction argument, it simply follows that all error vectors $(\theta_k - \hat{\theta})_{k \geq 0}$ remain parallel to each other while their norm contracts at a rate of $\frac{q-2}{q-1}$. This is a key point that is used in the result for Newton's method in Theorem 3. In the convergence analysis of BFGS (stated in the proof of Theorem 1), we use a similar argument. While we cannot guarantee that the Hessian approximation matrix in the BFGS method matches the exact Hessian, we demonstrate that it can inherit this property, maintaining all error vectors $(\theta_k - \hat{\theta})_{k \geq 0}$ as parallel to each other. Additionally, we show that the norm is shrinking at a factor of $r_k < 1$, which is always larger than $\frac{q-2}{q-1}$, yet remains independent of the problem's condition number or dimensions. In the following theorem, we show that for $q \geq 4$, the linear rate contraction factors $\{r_k\}_{k=0}^\infty$ also converge linearly to a fixed point contraction factor $r_*$ determined by the parameter $q$. The proof is available in Appendix A.2.

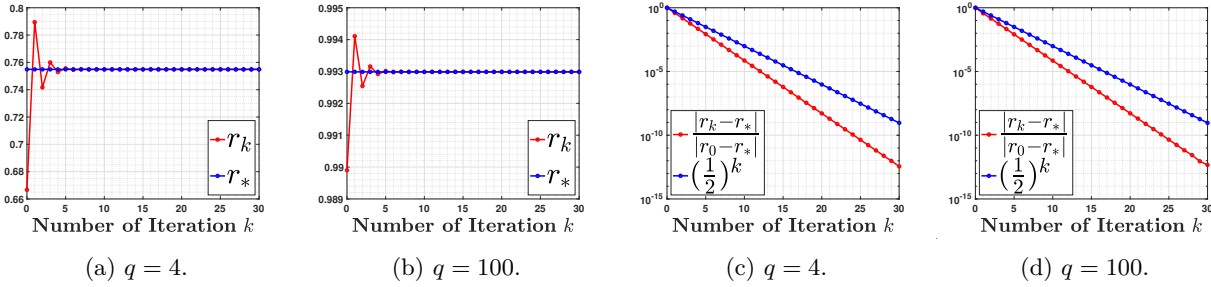

Figure 1: Convergence of factors $\{r_k\}_{k=0}^{\infty}$ to $r_*$.

**Theorem 2.** *Consider the linear convergence factors $\{r_k\}_{k=0}^{\infty}$ defined in (12) from Theorem 1. If $q \geq 4$ and $q \in \mathbb{Z}$, then the sequence $\{r_k\}_{k=0}^{\infty}$ converges to $r_* \in (0, 1)$ that is determined by the equation*

$$r_*^{q-1} + r_*^{q-2} = 1, \tag{13}$$

*and the rate of convergence is linear with a contraction factor that is at most $1/2$, i.e.,*

$$|r_k - r_*| \leq (1/2)^k |r_0 - r_*|, \qquad \forall k \geq 0. \tag{14}$$

Based on Theorem 2, the iterates of BFGS eventually converge to the optimal solution at the linear rate of $r_*$ determined by (13). Specifically, the factors $\{r_k\}_{k=0}^{\infty}$ converge to the fixed point $r_*$ at a linear rate with the contraction factor of $1/2$. Further, the linear convergence factors $\{r_k\}_{k=0}^{\infty}$ and their limit $r_*$ are all only determined by $q$, and they are independent of the dimension $d$ and the condition number $\kappa_A$ of the matrix $A$. Hence, the performance of BFGS is not influenced by high-dimensional or ill-conditioned problems. This result is independently important from an optimization point of view, as it provides the first global linear convergence of BFGS without line-search for a setting that is not strictly or strongly convex, and interestingly the constant of linear convergence is independent of dimension or condition number.

We illustrate the convergence of factors $\{r_k\}_{k=0}^{\infty}$ to the fixed point $r_*$ in Figure 1 for $q = 4$ and $q = 100$. In plots (a) and (b), we observe that $r_k$ becomes close to $r_*$ after only 5 iterations. Hence, the linear convergence rate of BFGS is approximately $r_*$ after only a few iterations. We further observe in plots (c) and (d) that the factors $\{r_k\}_{k=0}^{\infty}$ converge to the fixed point $r_*$ at a linear rate upper bounded by $1/2$. Note that $r_*$ is the solution of (13). These plots verify our results in Theorem 2.

**Remark 2.** *The cost of computing the initial Hessian inverse approximation $H_0 = \nabla^2 f(\theta_0)^{-1}$ is $\mathcal{O}(d^3)$, but this cost is only required for the first iteration, and it is not required for $k \geq 1$ as for those iterates we update the Hessian inverse approximation matrix $H_k$ based on the update in (4) at a cost of $\mathcal{O}(d^2)$.*

**Remark 3.** *Although the vanilla Gradient Descent (GD) method converges at a sub-linear rate when applied to problem (10), it has been shown that other first-order methods, such as Mirror Descent with a distance-generating function $h(x) = \frac{1}{q+2}|x|^{q+2} + \frac{1}{2}|x|^2$ can solve problem (10) at a linear rate (refer to Lu et al. (2018)). However, the linear convergence rate of Mirror Descent is dependent on the condition number $\kappa$ of the problem, characterized by the rate $\left(1 - \frac{1}{\kappa}\right)$. In contrast, as demonstrated by Theorems 1 and 2, the linear convergence rate of BFGS is independent of the problem's condition number. Consequently, in settings with low SNR, which can be ill-conditioned, we anticipate a faster convergence rate for BFGS compared to Mirror Descent. We illustrate this point in our numerical experiments presented in Figure 2.*

### 4.1 Comparison with Newton's method

Next, we compare the convergence results of BFGS for solving problem (10) with the one for Newton's method. The following theorem characterizes the global linear convergence of Newton's method with unit step size applied to the objective function in (10).

**Theorem 3.** *Consider applying Newton's method to optimization problem (10) and suppose Assumptions 1 and 2 hold. Moreover, suppose the step size is $\eta_k = 1$ for any $k \geq 0$. Then, the iterates of Newton's method*

*converge to the optimal solution $\hat{\theta}$ at a linear rate of*

$$\|\theta_k - \hat{\theta}\| = \frac{q-2}{q-1}\|\theta_{k-1} - \hat{\theta}\|, \quad \forall k \geq 1. \tag{15}$$

*Moreover, this linear convergence rate $\frac{q-2}{q-1}$ is smaller than the fixed point $r_*$ defined in (13) of the BFGS quasi-Newton method, i.e., $\frac{q-2}{q-1} < r_* < \frac{2q-3}{2q-2}$ for all $q \geq 4$.*

The proof is available in Appendix A.3. The convergence results of Newton's method are also global without any line search method, and the linear rate $\frac{q-2}{q-1}$ is independent of dimension $d$ and condition number $\kappa_A$. Furthermore, the condition $\frac{q-2}{q-1} < r_*$ implies that iterates of Newton's method converge faster than BFGS, but the gap is not substantial as $r_* < \frac{2q-3}{2q-2}$. On the other hand, the computational cost per iteration of Newton's method is $\mathcal{O}(d^3)$ which is worse than the $\mathcal{O}(d^2)$ of BFGS.

Moving back to our main problem, one important implication of the above convergence results is that in the low SNR setting the iterates of BFGS converge linearly to the optimal solution of the population loss function, while the contraction coefficient of BFGS is comparable to that of Newton's method which is $(2p-2)/(2p-1)$. For instance, for $p = 2, 3, 5, 10$, the linear rate contraction factor of Newton's method are $0.667, 0.8, 0.889, 0.947$ and the approximate linear rate contraction factor of BFGS denoted by $r_*$ are $0.755, 0.857, 0.922, 0.963$, respectively.

## 5 Convergence analysis in the low SNR regime: Finite sample setting

Thus far, we have demonstrated that the BFGS iterates converge linearly to the true parameter $\theta^*$ when minimizing the population loss function $\mathcal{L}$ of the GLM in (9) in the low SNR regime. In this section, we study the statistical behavior of the BFGS iterates for the finite sample case by leveraging the insights developed in the previous section about the convergence rate of BFGS in the infinite sample case, i.e., population loss. More precisely, we focus on the application of BFGS for solving the least-square loss function $\mathcal{L}_n$ defined in (7) for the low SNR setting. The iterates of BFGS in this case follow the update rule

$$\theta_{k+1}^n = \theta_k^n - \eta_k H_k^n \nabla \mathcal{L}_n(\theta_k^n), \tag{16}$$

where $H_k^n$ is updated using the gradient information of the loss $\mathcal{L}_n$ by the BFGS rule.

We next show that the BFGS iterates (16) $\{\theta_k^n\}_{k\geq 0}$ converge to the final statistical radius within a logarithmic number of iterations under the low SNR regime of the GLMs. To prove this claim, we track the difference between the iterates $\{\theta_k^n\}_{k\geq 0}$ generated based on the empirical loss and the iterates $\{\theta_k\}_{k\geq 0}$ generated according to the population loss. Assuming that they both start from the same initialization $\theta_0$, with the concentration of the gradient $\|\nabla \mathcal{L}_n(\theta) - \nabla \mathcal{L}(\theta)\|$ and the Hessian $\|\nabla^2 \mathcal{L}_n(\theta) - \nabla^2 \mathcal{L}(\theta)\|_{\mathrm{op}}$ from Mou et al. (2019); Ren et al. (2022b) we control the deviation between these two sequences. Using this bound and the convergence results of the iterates generated based on the population loss discussed in the previous section, we prove the following result for the finite sample setting. The proof is available in Appendix A.5.

**Theorem 4.** *Consider the low SNR regime of the GLM in (6) namely, $\|\theta^*\| \leq C_1(d/n)^{1/(2p)}$. Apply the BFGS method to the empirical loss (7) with the initial point $\theta_0^n$, where $\theta_0^n \in \mathbb{B}(\theta^*, r)$ for some $r > 0$, the initial Hessian inverse approximation matrix as $H_0 = \nabla^2 \mathcal{L}_n(\theta_0^n)^{-1}$, where $\nabla^2 \mathcal{L}_n(\theta_0^n)$ is positive definite and step size $\eta_k = 1$. For any failure probability $\delta \in (0, 1)$, if the number of samples is $n \geq C_2(d\log(d/\delta))^{2p}$, and the number of iterations satisfies $T \geq C_3 \log(n/d(\log(1/\delta)))$, then with probability $1 - \delta$, we have*

$$\min_{t \in [T]} \|\theta_t^n - \theta^*\| \leq C_4 \left(\frac{d\log(1/\delta)}{n}\right)^{\frac{1}{2p+2}}, \tag{17}$$

*where $C_1$, $C_2$, $C_3$, and $C_4$ are constants independent of $n$ and $d$.*

Our analysis hinges on linking the gradient and Hessian of the empirical loss to the population loss, subsequently demonstrating a convergence rate for the empirical loss based on the linear convergence established in Section 4

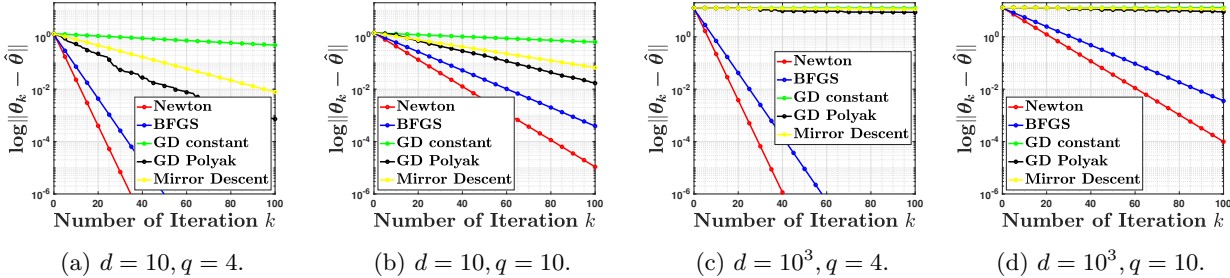

(a) $d = 10, q = 4.$      (b) $d = 10, q = 10.$      (c) $d = 10^3, q = 4.$      (d) $d = 10^3, q = 10.$

Figure 2: Convergence of Newton's method, BFGS, GD with constant step size, GD with Polyak step size and mirror descent with constant step size for different values of $d$ and $q$. In plot (a), $m = 100$ and $\eta = 10^{-4}$. In plot (b), $m = 100$ and $\eta = 10^{-8}$. In plot (c), $m = 2000$ and $\eta = 10^{-12}$. In plot (d), $m = 2000$ and $\eta = 10^{-15}$.

for the population loss. More details can be found in Appendix A.5. Theorem 4 shows that BFGS achieves an estimation accuracy of $\mathcal{O}((d/n)^{\frac{1}{2p+2}})$ in $O(\log n)$ iterations, which is substantialy faster than GD that requires $O(n^{\frac{p-1}{p}})$ iteations to achieve the same guarantee as shown in (Ren et al., 2022a). A few comments about Theorem 4 are in order.

**Comparison to GD, GD with Polyak step size, and Newton's method:** Theorem 4 indicates that under the low SNR regime, the BFGS iterates reach the final statistical radius $\mathcal{O}(n^{-1/(2p+2)})$ within the true parameter $\theta^*$ after $\mathcal{O}(\log(n))$ number of iterations. The statistical radius is slightly worse than the optimal statistical radius $O(n^{-1/(2p)})$. However, we conjecture that this is due to the proof technique and BFGS can still reach the optimal $O(n^{-1/(2p)})$ in practice. In our experiments, in the next section, we observe that when $d = 4$ and $p = 2$, the statistical radius of BFGS is closer to the optimal radius of $O(n^{-1/4})$ instead of $O(n^{-1/6})$ suggested by our analysis. We leave an improvement of the statistical analysis as the future work. On the other hand, the overall iteration complexity of BFGS, which is $\mathcal{O}(\log(n))$, is indeed better than the polynomial number of iterations of GD, which is at the order of $\mathcal{O}(n^{(p-1)/p})$ (Corollary 3 in (Ho et al., 2020)).

Moreover, the complexity of BFGS is better than the one for GD with Polyak step size which is $\mathcal{O}(\kappa \log(n))$ iterations (Corollary 1 in (Ren et al., 2022a)), where $\kappa$ is the condition number of the covariance matrix $\Sigma$. Note that while the iteration complexity of BFGS is comparable to that of GD with Polyak step size in terms of the sample size, the BFGS overcomes the need to approximate the optimal value of the sample least-square loss $\mathcal{L}_n$, which can be unstable in practice, and also removes the dependency on the condition number that appears in the complexity bound of GD with Polyak step size. A similar conclusion also holds for mirror descent with a distance-generating function $h(x) = \frac{1}{q+2}|x|^{q+2} + \frac{1}{2}|x|^2$, as its linear convergence rate has a contraction factor of $(1 - 1/\kappa)$, leading to an overall iteration complexity of $\mathcal{O}(\kappa \log(n))$.

Finally, the iteration complexity of BFGS is comparable to the $\mathcal{O}(\log(n))$ of Newton's method (Corollary 3 in (Ho et al., 2020)), while per iteration cost of BFGS is substantially lower than Newton's method.

**On the minimum number of iterations:** The results in Theorem 4 involve the minimum number of iterations, namely, this result holds for some $1 \le t \le T$. It suggests that the BFGS iterates may diverge after they reach the final statistical radius. As highlighted in (Ho et al., 2020), such instability behavior of BFGS is inherent to fast and unstable methods. While it may sound limited, this issue can be handled via an early stopping scheme using the cross-validation approaches. We illustrate such early stopping of the BFGS iterates for the low SNR regime in Figure 3.

## 6 Numerical experiments

Our experiments are divided into two sections: the first focuses on iterative methods' behavior on the population loss of GLMs with polynomial link functions, and the second examines the finite sample setting.

**Experiments for the population loss function.** In this section, we compare the performance of Newton's method, BFGS, GD with constant step size, GD with Polyak step size, and and mirror descent with constant

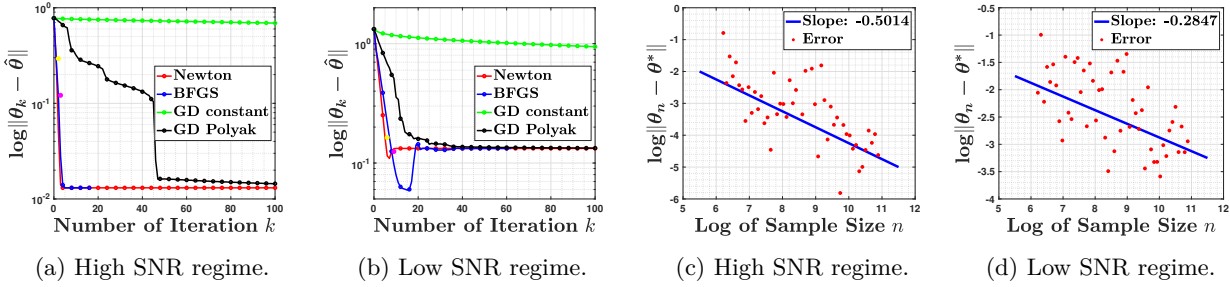

(a) High SNR regime.  (b) Low SNR regime.  (c) High SNR regime.  (d) Low SNR regime.

Figure 3: Convergence of different methods when $d = 4$ in high SNR (a) and low SNR (b) regimes. Illustration of the statistical radius of BFGS in high SNR (c) and low SNR (d) regimes.

step size and distance generating function $h(x) = \frac{1}{q+2}\|x\|^{q+2} + \frac{1}{2}\|x\|^2$ applied to (10) which corresponds to the population loss. We choose different values of parameter $m$, dimension $d$ and the exponential parameter $q$ in (10). We generate a random matrix $A \in \mathbb{R}^{m \times d}$ and a random vector $\hat{\theta} \in \mathbb{R}^d$, and compute the vector $b = A\hat{\theta} \in \mathbb{R}^d$. The initial point $\theta_0 \in \mathbb{R}^d$ is also generated randomly. To properly select the steps for GD with a fixed step size $\eta$, we employed a manual grid search across the following values $[10^{-10}, 10^{-9}, ..., 10^{-1}, 1]$ and selected the value that yielded the best performance for each specific problem. In our plots, we present the logarithm of error $\|\theta_k - \hat{\theta}\|$ versus the number of iteration $k$ for different algorithms. All the values of different parameters $m$, $d$, $q$ and $\eta$ are mentioned in the caption of the plots.

In Figure 2, we observe that GD with constant step converges slowly due to its sub-linear convergence rate. The performance of GD with Polyak step size is also poor when dimension is large or the parameter $q$ is huge. This is due to the fact that as dimension increases the problem becomes more ill-conditioned and hence the linear convergence contraction factor approaches 1. We observe that both Newton's method and BFGS generate iterations with linear convergence rates, and their linear convergence rates are only affected by the parameter $q$, i.e., the dimension $d$ has no impact over the performance of BFGS and Newton's method. Although the convergence speed of Newton's method is faster than BFGS, their gap is not significantly large as we expected from our theoretical results in Section 4. We also observe that mirror descent outperforms GD, but its performance is not as good as the BFGS method.

**Experiments for the empirical loss function.** We next study the statistical and computational complexities of BFGS on the empirical loss. In our experiments, we first consider the case that $d = 4$ and the power of the link function is $p = 2$, namely, we consider the multivariate setting of the phase retrieval problem. The data is generated by first sampling the inputs according to $\{X_i\}_{i=1}^n \sim \mathcal{N}(0, \text{diag}(\sigma_1^2, \cdots, \sigma_4^2))$ where $\sigma_k = (0.5)^{k-1}$, and then generating their labels based on $Y_i = (X_i^\top \theta^*)^2 + \zeta_i$ where $\{\zeta_i\}_{i=1}^n$ are i.i.d. samples from $\mathcal{N}(0, 0.01)$. In the low SNR regime, we set $\theta^* = 0$, and in the high SNR regime we select $\theta^*$ uniformly at random from the unit sphere. Further, for GD, we set the step size as $\eta = 0.1$, while for Newton's method and BFGS, we use the unit step size $\eta = 1$.

In plots (a) and (b) of Figure 3, we consider the setting that the sample size is $n = 10^4$, and we run GD, GD with Polyak step size, BFGS, and Newton's method to find the optimal solution of the sample least-square loss $\mathcal{L}_n$. Furthermore, for both Newton's method and the BFGS algorithm, due to their instability, we also perform cross-validation to choose their early stopping. In particular, we split the data into training and the test sets. The training set consists of 90% of the data while the test set has 10% of the data. The yellow points in plots (a) and (b) of Figure 3 show the iterates of BFGS and Newton, respectively, with the minimum validation loss. As we observe, under the low SNR regime, the iterates of GD with Polyak step size, BFGS and Newton's method converge geometrically fast to the final statistical radius while those of the GD converge slowly to that radius. Under the high SNR regime, the iterates of all of these methods converge geometrically fast to the final statistical radius. The faster convergence of GD with Polyak step size over GD is due to the optimal Polyak step size, while the faster convergence of BFGS and Newton's method over GD is due to their independence on the problem condition number. Finally, in plots (c) and (d) of Figure 3, we run BFGS when the sample size is ranging from $10^2$ to $10^4$ to empirically verify the statistical radius of

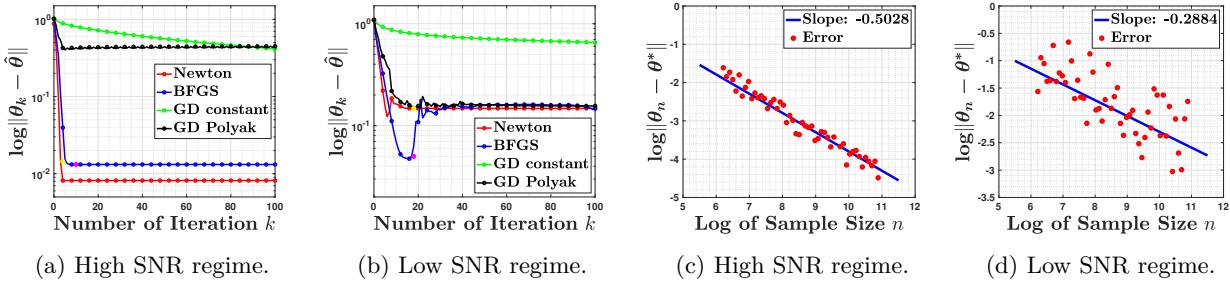

(a) High SNR regime.    (b) Low SNR regime.    (c) High SNR regime.    (d) Low SNR regime.

Figure 4: Convergence of different methods $d = 50$ in high SNR (a) and low SNR (b) regimes. Statistical radius of BFGS in high SNR (c) and low SNR (d) settings.

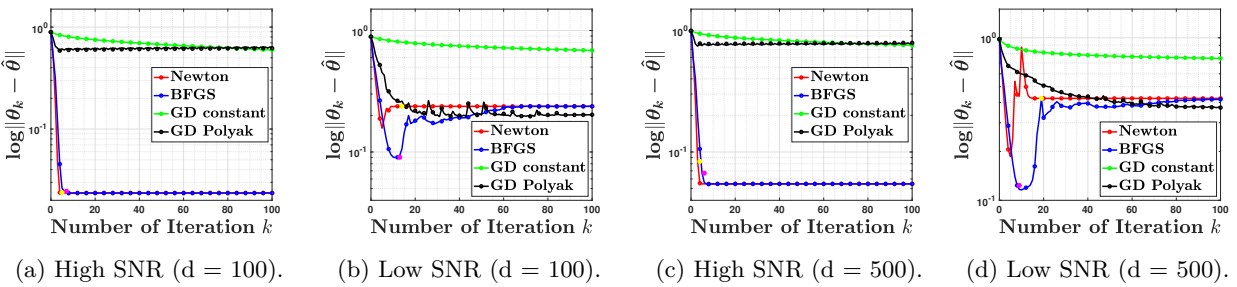

(a) High SNR (d = 100).    (b) Low SNR (d = 100).    (c) High SNR (d = 500).    (d) Low SNR (d = 500).

Figure 5: Convergence of different methods with $d = 100$ for high SNR regime are shown in (a) and low SNR regime in (b). Convergence of different methods with $d = 500$ for high SNR regime are shown in (c) and low SNR regime in (d).

these methods. As indicated in the plots of that figure, under the high SNR regime, the BFGS has statistical radius is $\mathcal{O}(n^{-1/2})$, while under the low SNR regime, its statistical radius becomes $\mathcal{O}(n^{-1/4})$.

To show that BFGS can also be applied to high dimension scenarios, we conduct additional experiments on the generalized linear model with input $d = 50, 100, 500$ and the power of link function $p = 2$. The inputs are generated by $\{X_i\}_{i=1}^n \sim \mathcal{N}(0, \mathrm{diag}(\sigma_1^2, \cdot, \sigma_d^2))$ where $\sigma_k = (0.96)^{k-1}$, and the remaining setting and hyper-parameters are set identical to the low dimension scenarios. The results are shown in Figure 4 and 5. As the results show, the performance of BFGS in high dimensional scenarios are nearly identical to the low dimensional scenarios.

## 7 Conclusions

In this paper, we analyzed the convergence rates of BFGS on both population and empirical loss functions of the generalized linear model in the low SNR regime. We showed that in this case, BFGS outperforms GD and performs similar to Newton's method in terms of iteration complexity, while it requires a lower per iteration computational complexity compared to Newton's method. We also provided experiments for both infinite and finite sample loss functions and showed that our empirical results are consistent with our theoretical findings. Perhaps one limitation of the BFGS method is that its computational cost is still not linear in the dimension and scales as $\mathcal{O}(d^2)$. One future research direction is to analyze some other iterative methods such as limited memory-BFGS (L-BFGS) which may be able to achieve a fast linear convergence rate in the low SNR setting, while its computational cost per iteration is $\mathcal{O}(d)$.

## Acknowledgements

The research of Qiujiang Jin and Aryan Mokhtari is partially supported by NSF Award CCF-2007668 and the NSF AI Institute for Foundations of Machine Learning. Tongzheng Ren and Naht Ho acknowledge support from the NSF IFML 2019844 and the NSF AI Institute for Foundations of Machine Learning.

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

## Appendix

## A    Proofs

**Lemma 1.** *Consider the objective function in* (10) *satisfying Assumption 1 and 2. Then, the inverse matrix of its Hessian* $\nabla^2 f(\theta)$ *can be expressed as*

$$\nabla^2 f(\theta)^{-1} = \frac{(A^\top A)^{-1}}{q\|A\theta - b\|^{q-2}} - \frac{(q-2)(\theta - \hat{\theta})(\theta - \hat{\theta})^\top}{q(q-1)\|A\theta - b\|^q}. \tag{18}$$

*Proof.* Notice that the Hessian of objective function (10) can be expressed as

$$\nabla^2 f(\theta) = q\|A\theta - b\|^{q-2} A^\top A + q(q-2)\|A\theta - b\|^{q-4} A^\top (A\theta - b)(A\theta - b)^\top A. \tag{19}$$

We use the Sherman–Morrison formula. Suppose that $X \in \mathbb{R}^{d \times d}$ is an invertible matrix and $a, b \in \mathbb{R}^d$ are two vectors satisfying that $1 + b^\top X^{-1} a \neq 0$. Then, the matrix $X + ab^\top$ is invertible and

$$(X + ab^\top)^{-1} = X^{-1} - \frac{X^{-1} ab^\top X^{-1}}{1 + b^\top X^{-1} a}.$$

Applying the Sherman–Morrison formula with $X = q\|A\theta - b\|^{q-2} A^\top A$, $a = q(q-2)\|A\theta - b\|^{q-4} A^\top (A\theta - b)$ and $b = A^\top (A\theta - b)$. Notice that $A^\top A$ is invertible, hence $X$ is invertible and

$$
\begin{aligned}
&1 + b^\top X^{-1} a \\
=\ & 1 + (A\theta - b)^\top A \frac{(A^\top A)^{-1}}{q\|A\theta - b\|^{q-2}} q(q-2)\|A\theta - b\|^{q-4} A^\top (A\theta - b) \\
=\ & 1 + (q-2)(A\theta - b)^\top A \frac{(A^\top A)^{-1} A^\top A(\theta - \hat{\theta})}{\|A\theta - b\|^2} \\
=\ & 1 + (q-2)\frac{(A\theta - b)^\top (A\theta - b)}{\|A\theta - b\|^2} \\
=\ & q - 1 \neq 0. \qquad \text{(Since } q \geq 4.\text{)}
\end{aligned}
$$

Therefore, we obtain that

$$
\begin{aligned}
&\nabla^2 f(\theta)^{-1} \\
=\ & \frac{(A^\top A)^{-1}}{q\|A\theta - b\|^{q-2}} - \frac{\frac{(A^\top A)^{-1}}{q\|A\theta - b\|^{q-2}} q(q-2)\|A\theta - b\|^{q-4} A^\top (A\theta - b)(A^\top (A\theta - b))^\top \frac{(A^\top A)^{-1}}{q\|A\theta - b\|^{q-2}}}{q - 1} \\
=\ & \frac{(A^\top A)^{-1}}{q\|A\theta - b\|^{q-2}} - \frac{(q-2)}{q(q-1)\|A\theta - b\|^q}(A^\top A)^{-1} AA^\top (\theta - \hat{\theta})(\theta - \hat{\theta})^\top AA^\top (A^\top A)^{-1} \\
=\ & \frac{(A^\top A)^{-1}}{q\|A\theta - b\|^{q-2}} - \frac{(q-2)(\theta - \hat{\theta})(\theta - \hat{\theta})^\top}{q(q-1)\|A\theta - b\|^q}.
\end{aligned} \tag{20}
$$

As a consequence, we obtain the conclusion of the lemma. $\qquad \square$

**Lemma 2. *Banach's Fixed-Point Theorem.*** *Consider the differentiable function $f : D \subset \mathbb{R} \to D \subset \mathbb{R}$. Suppose that there exists $C \in (0, 1)$ such that $|f'(x)| \leq C$ for any $x \in D$. Now let $x_0 \in D$ be arbitrary and define the sequence $\{x_k\}_{k=0}^\infty$ as*

$$x_{k+1} = f(x_k), \qquad \forall k \geq 0. \tag{21}$$

*Then, the sequence $\{x_k\}_{k=0}^\infty$ converges to the unique fixed point $x_*$ defined as*

$$x_* = f(x_*), \tag{22}$$

*with linear convergence rate of*

$$|x_k - x_*| \leq C^k |x_0 - x_*|, \qquad \forall k \geq 0. \tag{23}$$

*Proof.* Check (Goebel & Kirk, 1990). $\qquad \square$

### A.1 Proof of Theorem 1

We use induction to prove the convergence results in (11) and (12). Note that $b = A\hat{\theta}$ by Assumption 1 and the gradient and Hessian of the objective function in (10) are explicitly given by

$$\nabla f(\theta) = q\|A\theta - b\|^{q-2}A^\top(A\theta - b) = q\|A\theta - b\|^{q-2}A^\top A(\theta - \hat{\theta}), \tag{24}$$

$$\nabla^2 f(\theta) = q\|A\theta - b\|^{q-2}A^\top A + q(q-2)\|A\theta - b\|^{q-4}A^\top(A\theta - b)(A\theta - b)^\top A. \tag{25}$$

Applying Lemma 1, we can obtain that

$$\nabla^2 f(\theta)^{-1} = \frac{(A^\top A)^{-1}}{q\|A\theta - b\|^{q-2}} - \frac{(q-2)(\theta - \hat{\theta})(\theta - \hat{\theta})^\top}{q(q-1)\|A\theta - b\|^q}. \tag{26}$$

First, we consider the initial iteration

$$\theta_1 = \theta_0 - H_0\nabla f(\theta_0) = \theta_0 - \nabla f(\theta_0)^{-1}\nabla f(\theta_0),$$

$$\theta_1 - \hat{\theta} = \theta_0 - \hat{\theta} - \nabla f(\theta_0)^{-1}\nabla f(\theta_0).$$

Notice that $b = A\hat{\theta}$ by Assumption 1 and

$$
\begin{aligned}
&\nabla^2 f(\theta_0)^{-1}\nabla f(\theta_0)\\
&= \left[\frac{(A^\top A)^{-1}}{q\|A\theta_0 - b\|^{q-2}} - \frac{(q-2)(\theta_0 - \hat{\theta})(\theta_0 - \hat{\theta})^\top}{q(q-1)\|A\theta_0 - b\|^q}\right]q\|A\theta_0 - b\|^{q-2}A^\top A(\theta_0 - \hat{\theta})\\
&= \theta_0 - \hat{\theta} - \frac{q-2}{q-1}\frac{(\theta_0 - \hat{\theta})^\top A^\top A(\theta_0 - \hat{\theta})}{\|A\theta_0 - b\|^2}(\theta_0 - \hat{\theta})\\
&= \theta_0 - \hat{\theta} - \frac{q-2}{q-1}\frac{(A\theta_0 - b)^\top(A\theta_0 - b)}{\|A\theta_0 - b\|^2}(\theta_0 - \hat{\theta})\\
&= \theta_0 - \hat{\theta} - \frac{q-2}{q-1}(\theta_0 - \hat{\theta}).
\end{aligned}
\tag{27}
$$

Therefore, we obtain that

$$\theta_1 - \hat{\theta} = \theta_0 - \hat{\theta} - \nabla f(\theta_0)^{-1}\nabla f(\theta_0) = \frac{q-2}{q-1}(\theta_0 - \hat{\theta}).$$

Condition (11) holds for $k = 1$ with $r_0 = \frac{q-2}{q-1}$. Now we assume that condition (11) holds for $k = t$ where $t \geq 1$, i.e.,

$$\theta_t - \hat{\theta} = r_{t-1}(\theta_{t-1} - \hat{\theta}). \tag{28}$$

Considering the condition $b = A\hat{\theta}$ in Assumption 1 and the condition in (28), we further have

$$A\theta_t - b = A(\theta_t - \hat{\theta}) = r_{t-1}A(\theta_{t-1} - \hat{\theta}) = r_{t-1}(A\theta_{t-1} - b),$$

which implies that

$$\nabla f(\theta_t) = qr_{t-1}^{q-1}\|A(\theta_{t-1} - \hat{\theta})\|^{q-2}A^\top A(\theta_{t-1} - \hat{\theta}). \tag{29}$$

We further show that the variable displacement and gradient difference can be written as

$$s_{t-1} = \theta_t - \theta_{t-1} = \theta_t - \hat{\theta} - \theta_{t-1} + \hat{\theta} = (r_{t-1} - 1)(\theta_{t-1} - \hat{\theta}),$$

and

$$u_{t-1} = \nabla f(\theta_t) - \nabla f(\theta_{t-1}) = q(r_{t-1}^{q-1} - 1)\|A(\theta_{t-1} - \hat{\theta})\|^{q-2}A^\top A(\theta_{t-1} - \hat{\theta}).$$

Considering these expressions, we can show that the rank-1 matrix in the update of BFGS $u_{t-1}s_{t-1}^\top$ is given by

$$u_{t-1}s_{t-1}^\top = q(r_{t-1}^{q-1} - 1)(r_{t-1} - 1)\|A(\theta_{t-1} - \hat{\theta})\|^{q-2}A^\top A(\theta_{t-1} - \hat{\theta})(\theta_{t-1} - \hat{\theta})^\top,$$

and the inner product $s_{t-1}^\top u_{t-1}$ can be written as

$$\begin{aligned} s_{t-1}^\top u_{t-1} &= q(r_{t-1}^{q-1} - 1)(r_{t-1} - 1)\|A(\theta_{t-1} - \hat\theta)\|^{q-2}(\theta_{t-1} - \hat\theta)^\top A^\top A(\theta_{t-1} - \hat\theta) \\ &= q(r_{t-1}^{q-1} - 1)(r_{t-1} - 1)\|A(\theta_{t-1} - \hat\theta)\|^q. \end{aligned}$$

These two expressions allow us to simplify the matrix $I - \frac{u_{t-1}s_{t-1}^\top}{s_{t-1}^\top u_{t-1}}$ in the update of BFGS as

$$I - \frac{u_{t-1}s_{t-1}^\top}{s_{t-1}^\top u_{t-1}} = I - \frac{A^\top A(\theta_{t-1} - \hat\theta)(\theta_{t-1} - \hat\theta)^\top}{\|A(\theta_{t-1} - \hat\theta)\|^2}. \tag{30}$$

An important property of the above matrix is that its null space is the set of the vectors that are parallel to $u_{t-1}$. Considering the expression for $u_{t-1}$, any vector that is parallel to $A^\top A(\theta_{t-1} - \hat\theta)$ is in the null space of the above matrix. We can easily observe that the gradient defined in (29) satisfies this condition and therefore

$$\begin{aligned} &\left(I - \frac{u_{t-1}s_{t-1}^\top}{s_{t-1}^\top u_{t-1}}\right)\nabla f(\theta_t) \\ =\ & qr_{t-1}^{q-1}\|A(\theta_{t-1} - \hat\theta)\|^{q-2}\left(I - \frac{A^\top A(\theta_{t-1} - \hat\theta)(\theta_{t-1} - \hat\theta)^\top}{\|A(\theta_{t-1} - \hat\theta)\|^2}\right)A^\top A(\theta_{t-1} - \hat\theta) \\ =\ & qr_{t-1}^{q-1}\|A(\theta_{t-1} - \hat\theta)\|^{q-2}\left(A^\top A(\theta_{t-1} - \hat\theta) - \frac{A^\top A(\theta_{t-1} - \hat\theta)\|A(\theta_{t-1} - \hat\theta)\|^2}{\|A(\theta_{t-1} - \hat\theta)\|^2}\right) \\ =\ & 0. \end{aligned} \tag{31}$$

This important observation shows that if the condition in (28) holds, then the BFGS descent direction $H_t\nabla f(\theta_t)$ can be simplified as

$$\begin{aligned} &H_t\nabla f(\theta_t) \\ =\ & \left(I - \frac{s_{t-1}u_{t-1}^\top}{s_{t-1}^\top u_{t-1}}\right)H_{t-1}\left(I - \frac{u_{t-1}s_{t-1}^\top}{s_{t-1}^\top u_{t-1}}\right)\nabla f(\theta_t) + \frac{s_{t-1}s_{t-1}^\top}{s_{t-1}^\top u_{t-1}}\nabla f(\theta_t) \\ =\ & \frac{s_{t-1}s_{t-1}^\top}{s_{t-1}^\top u_{t-1}}\nabla f(\theta_t) \\ =\ & \frac{(r_{t-1} - 1)^2(\theta_{t-1} - \hat\theta)(\theta_{t-1} - \hat\theta)^\top}{q(r_{t-1}^{q-1} - 1)(r_{t-1} - 1)\|A(\theta_{t-1} - \hat\theta)\|^q}qr_{t-1}^{q-1}\|A(\theta_{t-1} - \hat\theta)\|^{q-2}A^\top A(\theta_{t-1} - \hat\theta) \\ =\ & \frac{1 - r_{t-1}}{1 - r_{t-1}^{q-1}}r_{t-1}^{q-1}(\theta_{t-1} - \hat\theta)\frac{\|A(\theta_{t-1} - \hat\theta)\|^{q-2}(\theta_{t-1} - \hat\theta)^\top A^\top A(\theta_{t-1} - \hat\theta)}{\|A(\theta_{t-1} - \hat\theta)\|^q} \\ =\ & \frac{1 - r_{t-1}}{1 - r_{t-1}^{q-1}}r_{t-1}^{q-1}(\theta_{t-1} - \hat\theta). \end{aligned} \tag{32}$$

This simplification implies that for the new iterate $\theta_{t+1}$, we have

$$\begin{aligned} \theta_{t+1} - \hat\theta &= \theta_t - H_t\nabla f(\theta_t) - \hat\theta = \theta_t - \hat\theta - \frac{1 - r_{t-1}}{1 - r_{t-1}^{q-1}}r_{t-1}^{q-1}\frac{(\theta_t - \hat\theta)}{r_{t-1}} \\ &= \frac{1 - r_{t-1}^{q-2}}{1 - r_{t-1}^{q-1}}(\theta_t - \hat\theta) = r_t(\theta_t - \hat\theta), \end{aligned} \tag{33}$$

where

$$r_t = \frac{1 - r_{t-1}^{q-2}}{1 - r_{t-1}^{q-1}}. \tag{34}$$

Therefore, we prove that condition (11) holds for $k = t + 1$. By induction, we prove the linear convergence results in (11) and (12).

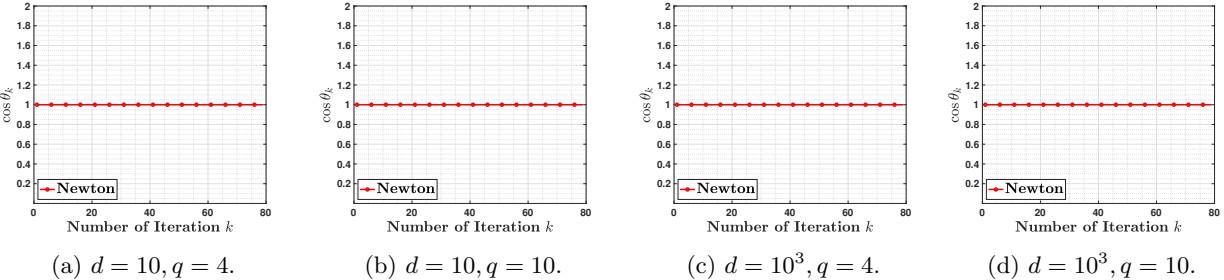

(a) $d = 10, q = 4$.     (b) $d = 10, q = 10$.     (c) $d = 10^3, q = 4$.     (d) $d = 10^3, q = 10$.

Figure 6: Values of $\cos\theta_k$ generated by Newton's method for different values of $d$ and $q$.

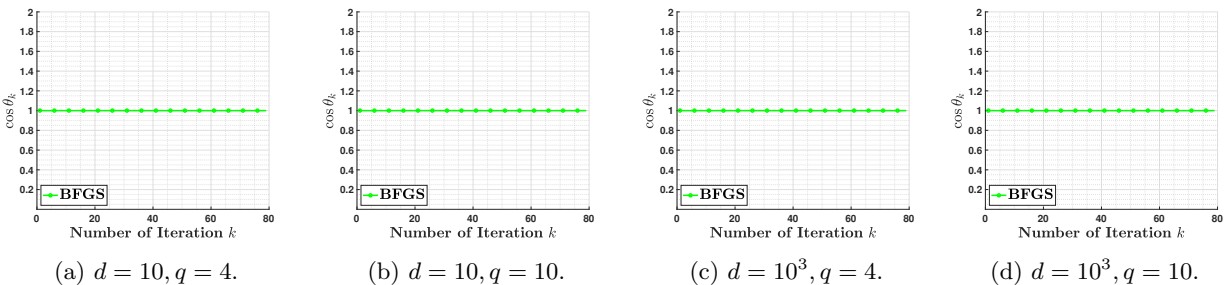

(a) $d = 10, q = 4$.     (b) $d = 10, q = 10$.     (c) $d = 10^3, q = 4$.     (d) $d = 10^3, q = 10$.

Figure 7: Values of $\cos\theta_k$ generated by the BFGS method for different values of $d$ and $q$.

One property of this convergence results is that the error vectors $\{\theta_k - \hat{\theta}\}_{k=0}^{\infty}$ are parallel to each other with the same direction as shown in (11). This indicates that the iterations $\{\theta_k\}_{k=0}^{\infty}$ converge to the optimal solution $\hat{\theta}$ along the same straight line defined by $\theta_0 - \hat{\theta}$. Only the length of each vector $\theta_k - \hat{\theta}$ reduces to zero with the linear convergence rates $\{r_k\}_{k=0}^{\infty}$ specified in (12) and the direction remains all the same. Notice that this is not a common property for BFGS and Newton's method. This property requires that the initial Hessian approximation matrix is $H_0 = \nabla^2 f(\theta_0)^{-1}$ and the convex problem must be quadratic problem (10). When the objective function is a general convex function or the initial Hessian approximation matrix is not $H_0 = \nabla^2 f(\theta_0)^{-1}$, there is no guarantee that the error vectors $\{\theta_k - \hat{\theta}\}_{k=0}^{\infty}$ are parallel to each other. To better visualize this property, we have plotted $\cos\theta_k = (\theta_k - \hat{\theta})^{\top}(\theta_0 - \hat{\theta})/\|\theta_k - \hat{\theta}\|\|\theta_0 - \hat{\theta}\|$ of $k \geq 1$ for both Newton's method and BFGS in Figures 6 and 7 with population loss function defined in (10). We observed that all values of $\cos\theta_k$ are one, which indicates that all vectors $\{\theta_k - \hat{\theta}\}_{k=0}^{\infty}$ are parallel to each other.

### A.2   Proof of Theorem 2

Recall that we have

$$r_0 = \frac{q-2}{q-1}, \qquad r_k = \frac{1 - r_{k-1}^{q-2}}{1 - r_{k-1}^{q-1}}, \qquad \forall k \geq 1. \tag{35}$$

Consider that $q \geq 4$ and define the function $g(r)$ as

$$g(r) := \frac{1 - r^{q-2}}{1 - r^{q-1}}, \qquad r \in [0, 1]. \tag{36}$$

Suppose that $r_* \in (0, 1)$ satisfying that $r_* = g(r_*)$, which is equivalent to

$$r_*^{q-1} + r_*^{q-2} = 1. \tag{37}$$

Notice that

$$g'(r) = \frac{(q-1)r^{q-2} - r^{2q-4} - (q-2)r^{q-3}}{(1 - r^{q-1})^2},$$

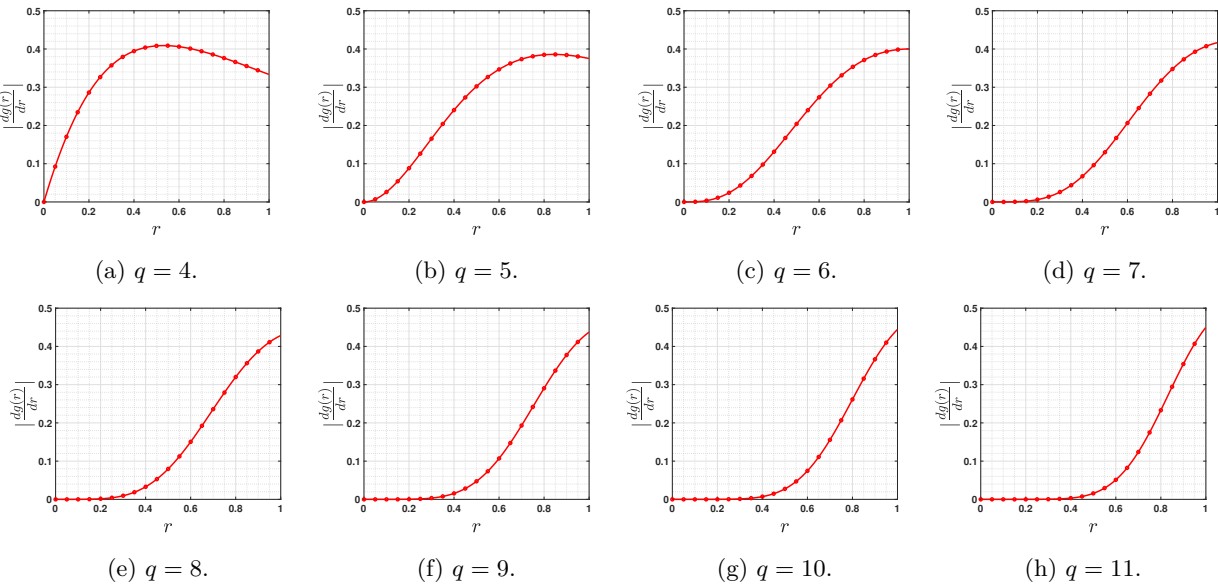

Figure 8: Plots of $|g'(r)|$ with $r \in [0, 1]$ for $4 \le q \le 11$.

and

$$
\begin{aligned}
&(q-1)r^{q-2} - r^{2q-4} - (q-2)r^{q-3} \\
=\ & r^{q-3}[(q-1)(r-1) - (r^{q-1} - 1)] \\
=\ & r^{q-3}(r-1)(q-1 - \frac{r^{q-1} - 1}{r-1}) \\
=\ & r^{q-3}(r-1)(q-1 - \sum_{i=0}^{q-2} r^i).
\end{aligned}
$$

Since $r \in [0, 1]$, we have that

$$
r^{q-3} \ge 0, \quad r-1 \le 0, \quad \sum_{i=0}^{q-2} r^i \le \sum_{i=0}^{q-2} 1 = q - 1.
$$

Therefore, we obtain that

$$
(q-1)r^{q-2} - r^{2q-4} - (q-2)r^{q-3} \le 0,
$$

and

$$
|g'(r)| = \frac{r^{2q-4} + (q-2)r^{q-3} - (q-1)r^{q-2}}{(1 - r^{q-1})^2}.
$$

Our target is to prove that for any $r \in [0, 1]$,

$$
|g'(r)| \le \frac{1}{2}.
$$

First, we present the plots of $|g'(r)|$ for $r \in [0, 1]$ with $4 \le q \le 11$ in Figure 8. We observe that for $4 \le q \le 11$, $|g'(r)| \le 1/2$ always holds.

Next, we prove that for $q \ge 12$ and any $r \in [0, 1]$, we have

$$
|g'(r)| = \frac{(q-1)r^{q-2} - r^{2q-4} - (q-2)r^{q-3}}{(1 - r^{q-1})^2} \le \frac{1}{2},
$$

which is equivalent to

$$r^{2q-2} - 2r^{2q-4} - 2r^{q-1} + 2(q-1)r^{q-2} - 2(q-2)r^{q-3} + 1 \geq 0, \qquad \forall r \in [0,1]. \tag{38}$$

Define the function $h(r)$ as

$$h(r) := r^{2q-2} - 2r^{2q-4} - 2r^{q-1} + 2(q-1)r^{q-2} - 2(q-2)r^{q-3} + 1. \tag{39}$$

We obtain that

$$\frac{dh(r)}{dr} = 2r^{q-4}h^{(1)}(r), \tag{40}$$

where

$$h^{(1)}(r) := (q-1)r^{q+1} - 2(q-2)r^{q-1} - (q-1)r^2 + (q-1)(q-2)r - (q-2)(q-3). \tag{41}$$

Hence, we have that

$$\frac{dh^{(1)}(r)}{dr} = (q-1)h^{(2)}(r), \tag{42}$$

where

$$h^{(2)}(r) := (q+1)r^q - 2(q-2)r^{q-2} - 2r + q - 2. \tag{43}$$

Therefore, we obtain that

$$\frac{dh^{(2)}(r)}{dr} = h^{(3)}(r) := (q+1)qr^{q-1} - 2(q-2)^2 r^{q-3} - 2, \tag{44}$$

and

$$\frac{dh^{(3)}(r)}{dr} = r^{q-4}h^{(4)}(r), \tag{45}$$

where

$$h^{(4)}(r) := q(q+1)(q-1)r^2 - 2(q-2)^2(q-3). \tag{46}$$

Now we define the function $l(q)$ as

$$\begin{aligned} l(q) := \quad & 2(q-2)^2(q-3) - q(q+1)(q-1) \\ = \quad & q^3 - 14q^2 + 33q - 24 \\ = \quad & q^2(q-14) + 33(q-1) + 9. \end{aligned}$$

We observe that for $q \geq 14$, we have $l(q) > 0$ and we calculate that $l(12) = 84 > 0$ and $l(13) = 236 > 0$. Hence, we obtain that $l(q) > 0$ for all $q \geq 12$, which indicates that for all $r \in [0,1]$,

$$r^2 \leq 1 < \frac{2(q-2)^2(q-3)}{q(q+1)(q-1)},$$

$$q(q+1)(q-1)r^2 - 2(q-2)^2(q-3) < 0.$$

Therefore, for all $r \in [0,1]$, $h^{(4)}(r)$ defined in (46) satisfies that $h^{(4)}(r) < 0$ and from (45) we know that $\frac{dh^{(3)}(r)}{dr} < 0$. Hence, $h^{(3)}(r)$ defined in (44) is decreasing function and $h^{(3)}(r) <= h^{(3)}(0) = -2 < 0$. We know that $\frac{dh^{(2)}(r)}{dr} = h^{(3)}(r) < 0$, which implies that $h^{(2)}(r)$ defined in (43) is decreasing function. So we have that $h^{(2)}(r) \geq h^{(2)}(1) = 1 > 0$. From (42) we know that $\frac{dh^{(1)}(r)}{dr} > 0$ and $h^{(1)}(r)$ defined in (41) is increasing function for $r \in [0,1]$. Hence, we get that $h^{(1)}(r) \leq h^{(1)}(1) = 0$ and from (40) we obtain that $h(r)$ defined in(39) is decreasing function for $r \in [0,1]$. Therefore, we have that $h(r) \geq h(1) = 0$ and condition (38) holds for all $r \in [0,1]$, which is equivalent to $|g'(r)| \leq 1/2$.

In summary, we proved that for any $q \geq 12$, we have $|g'(r)| \leq 1/2$. Combining this with the results from Figure 8, we obtain that $|g'(r)| \leq 1/2$ holds for all $q \geq 4$. Applying Banach's Fixed-Point Theorem from Lemma 2, we prove the final conclusion (14).

### A.3 Proof of Theorem 3

Notice that the gradient and the Hessian of the objective function (10) can be expressed as

$$\nabla f(\theta) = q\|A\theta - b\|^{q-2}A^\top(A\theta - b) = q\|A\theta - b\|^{q-2}A^\top A(\theta - \hat{\theta}),$$

$$\nabla^2 f(\theta) = q\|A\theta - b\|^{q-2}A^\top A + q(q-2)\|A\theta - b\|^{q-4}A^\top(A\theta - b)(A\theta - b)^\top A.$$

Applying Lemma 1, we can obtain that

$$\nabla^2 f(\theta)^{-1} = \frac{(A^\top A)^{-1}}{q\|A\theta - b\|^{q-2}} - \frac{(q-2)(\theta - \hat{\theta})(\theta - \hat{\theta})^\top}{q(q-1)\|A\theta - b\|^q}.$$

Hence, we have that for any $k \geq 1$,

$$\theta_k = \theta_{k-1} - \nabla^2 f(\theta_{k-1})^{-1}\nabla f(\theta_{k-1}),$$

$$\theta_k - \hat{\theta} = \theta_{k-1} - \hat{\theta} - \nabla^2 f(\theta_{k-1})^{-1}\nabla f(\theta_{k-1}).$$

Notice that $b = A\hat{\theta}$ by Assumption 1 and

$$
\begin{aligned}
&\nabla^2 f(\theta_{k-1})^{-1}\nabla f(\theta_{k-1}) \\
=\ & [\frac{(A^\top A)^{-1}}{q\|A\theta_{k-1} - b\|^{q-2}} - \frac{(q-2)(\theta_{k-1} - \hat{\theta})(\theta_{k-1} - \hat{\theta})^\top}{q(q-1)\|A\theta_{k-1} - b\|^q}]q\|A\theta - b\|^{q-2}A^\top A(\theta_{k-1} - \hat{\theta}) \\
=\ & \theta_{k-1} - \hat{\theta} - \frac{q-2}{q-1}\frac{(\theta_0 - \hat{\theta})^\top A^\top A(\theta_{k-1} - \hat{\theta})}{\|A\theta_{k-1} - b\|^2}(\theta_{k-1} - \hat{\theta}) \\
=\ & \theta_{k-1} - \hat{\theta} - \frac{q-2}{q-1}\frac{(A\theta_{k-1} - b)^\top(A\theta_{k-1} - b)}{\|A\theta_{k-1} - b\|^2}(\theta_{k-1} - \hat{\theta}) \\
=\ & \theta_{k-1} - \hat{\theta} - \frac{q-2}{q-1}(\theta_{k-1} - \hat{\theta}).
\end{aligned}
$$

Therefore, we prove the conclusion that for any $k \geq 1$,

$$\theta_k - \hat{\theta} = \theta_{k-1} - \hat{\theta} - \nabla^2 f(\theta_{k-1})^{-1}\nabla f(\theta_{k-1}) = \frac{q-2}{q-1}(\theta_{k-1} - \hat{\theta}).$$

We observe that the iterations generated by Newton's method also satisfy the parallel property, i.e., all vectors $\{\theta_k - \hat{\theta}\}_{k=0}^\infty$ are parallel to each other with the same direction.

Notice that the function $h(r) = r^{q-1} + r^{q-2}$ is strictly increasing and $h(\frac{q-2}{q-1}) < 1$, $h(r_*) = 1$ as well as $h(\frac{2q-3}{2q-2}) > 1$. Hence, we know that $\frac{q-2}{q-1} < r_* < \frac{2q-3}{2q-2}$.

### A.4 Elaboration of Remark 1

**For the ease of presentation, we use $C_p$ to denote any constant that is independent of $d$, $n$, and $C_p$ can be varied case by case to simplify the proof.** From Mou et al. (2019) and Lemma 6 of Ren et al. (2022b), we have that, as long as $n = \Omega((d\log d/\delta)^{2p})$, we have the following two uniform concentration results holding with probability $1 - \delta$:

$$
\begin{aligned}
\sup_{\theta \in \mathbb{B}(\theta^*, r)} \|\nabla\mathcal{L}_n(\theta)) - \nabla\mathcal{L}(\theta))\| &\leq C_p(\|\theta^*\| + r)^{p-1}\sqrt{d\log(1/\delta)/n}, \\
\sup_{\theta \in \mathbb{B}(\theta^*, r)} \|\nabla^2\mathcal{L}_n(\theta)) - \nabla^2\mathcal{L}(\theta))\| &\leq C_p(\|\theta^*\| + r)^{p-2}\sqrt{d\log(1/\delta)/n}.
\end{aligned}
\tag{47}
$$

Notice that we have

$$
\begin{aligned}
\mathcal{L}(\theta) &= \mathbb{E}[(Y - (X^\top\theta)^p)^2] = \mathbb{E}[((X^\top\theta^*)^p + \zeta - (X^\top\theta)^p)^2] \\
&= \mathbb{E}[((X^\top\theta^*)^p - (X^\top\theta)^p)^2] + \mathbb{E}[2\zeta((X^\top\theta^*)^p - (X^\top\theta)^p)] + \mathbb{E}[\zeta^2] \\
&= \mathbb{E}[((X^\top\theta^*)^p - (X^\top\theta)^p)^2] + \sigma^2,
\end{aligned}
$$

where we use the fact that $\zeta$ is independent of $X$ and $\mathbb{E}[\zeta] = 0$, $\mathbb{E}[\zeta^2] = \sigma^2$. Hence, we have that

$$\nabla\mathcal{L}(\theta) = 2p\mathbb{E}[((X^\top\theta)^p - (X^\top\theta^*)^p)(X^\top\theta)^{p-1}X]. \tag{48}$$

$$\nabla^2\mathcal{L}(\theta) = 2p^2\mathbb{E}[(X^\top\theta)^{2p-2}XX^\top] + 2p(p-1)\mathbb{E}[((X^\top\theta)^p - (X^\top\theta^*)^p)(X^\top\theta)^{p-2})XX^\top]. \tag{49}$$

We denote $\mathcal{L}^0$ as the population loss function with respect to the assumption of $\theta^* = 0$. Therefore, we have

$$\mathcal{L}^0(\theta) = \mathbb{E}[(X^\top\theta)^{2p}] + \sigma^2. \tag{50}$$

$$\nabla\mathcal{L}^0(\theta) = 2p\mathbb{E}[(X^\top\theta)^{2p-1}X]. \tag{51}$$

$$\nabla^2\mathcal{L}^0(\theta) = 2p(2p-1)\mathbb{E}[(X^\top\theta)^{2p-2}XX^\top]. \tag{52}$$

Hence, we have

$$\|\nabla\mathcal{L}(\theta) - \nabla\mathcal{L}^0(\theta)\| = C_p\mathbb{E}[(X^\top\theta^*)^p(X^\top\theta)^{p-1}X] \le C_p\mathbb{E}[\|X\|^{2p}]\|\theta^*\|^p\|\theta\|^{p-1}.$$

Recall that $X$ is a Gaussian or sub-Gaussian random variable with $\mathbb{E}[\|X\|^{2p}] < +\infty$. For any $\theta$, we ahve that $\|\theta\| \le \|\theta^*\| + \|\theta - \theta^*\|$ and in the low SNR regime, we have $\|\theta^*\| \le C_1(\frac{d}{n})^{\frac{1}{2p}}$. Hence, we have

$$\sup_{\theta\in\mathbb{B}(\theta^*,r)} \|\nabla\mathcal{L}(\theta) - \nabla\mathcal{L}^0(\theta)\| \le C_p(\|\theta^*\| + r)^{p-1}\sqrt{d\log(1/\delta)/n}. \tag{53}$$

Similarly, we have that

$$\|\nabla^2\mathcal{L}(\theta) - \nabla^2\mathcal{L}^0(\theta)\| = C_p\mathbb{E}[(X^\top\theta^*)^p(X^\top\theta)^{p-2}XX^\top] \le C_p\mathbb{E}[\|X\|^{2p}]\|\theta^*\|^p\|\theta\|^{p-2}.$$

$$\sup_{\theta\in\mathbb{B}(\theta^*,r)} \|\nabla^2\mathcal{L}(\theta) - \nabla^2\mathcal{L}^0(\theta)\| \le C_p(\|\theta^*\| + r)^{p-2}\sqrt{d\log(1/\delta)/n}. \tag{54}$$

Leveraging (47), (53) and (54), we obtain that

$$\sup_{\theta\in\mathbb{B}(\theta^*,r)} \|\nabla\mathcal{L}_n(\theta)) - \nabla\mathcal{L}^0(\theta))\| \le C_p(\|\theta^*\| + r)^{p-1}\sqrt{d\log(1/\delta)/n},$$
$$\sup_{\theta\in\mathbb{B}(\theta^*,r)} \|\nabla^2\mathcal{L}_n(\theta)) - \nabla^2\mathcal{L}^0(\theta))\| \le C_p(\|\theta^*\| + r)^{p-2}\sqrt{d\log(1/\delta)/n}. \tag{55}$$

This explains the Remark 1 of assumption $\theta^* = 0$ in section 4. The errors between gradients and Hessians of the population loss with $\theta^* = 0$ and $\|\theta^*\| \le C_1(d/n)^{1/(2p)}$ are upper bounded by the corresponding statistical errors between the population loss (8) and the empirical loss (7) in the low SNR regime, respectively. Hence, $\mathcal{L}^0$ and $\mathcal{L}$ can be treated as equivalent.

## A.5 Proof of Theorem 4

In this section, we present the proof of Theorem 4. We use $\mathcal{L}$ and $\mathcal{L}_n$ to denote the population objective and empirical objective. Also, as discussed in the previous section, we use $\mathcal{L}^0$ to refer to the population loss when the optimal solution is zero $\theta^* = 0$. **For the ease of presentation, we use $C_p$ to denote any constant that is independent of $d$, $n$, and $C_p$ can be varied case by case to simplify the proof.** To prove the result for the iterates generated by the finite sample loss function, we control the gap between the gradient of $\mathcal{L}^0$ and $\mathcal{L}_n$. As discussed previously in (55) from A.4, we can show that in the low SNR setting, we have

$$\sup_{\theta\in\mathbb{B}(\theta^*,r)} \|\nabla\mathcal{L}_n(\theta)) - \nabla\mathcal{L}^0(\theta))\| \le C_p(\|\theta^*\| + r)^{p-1}\sqrt{d\log(1/\delta)/n}, \tag{56}$$

$$\sup_{\theta\in\mathbb{B}(\theta^*,r)} \|\nabla^2\mathcal{L}_n(\theta)) - \nabla^2\mathcal{L}^0(\theta))\| \le C_p(\|\theta^*\| + r)^{p-2}\sqrt{d\log(1/\delta)/n}. \tag{57}$$

We further can show that $\|\theta^*\| \leq \|\theta - \theta^*\|$ for any $\theta$. If this does not hold, then we have $\|\theta - \theta^*\| < \|\theta^*\| \leq C_1(\frac{d}{n})^{\frac{1}{2p}}$ by the definition of the low SNR regime, which indicates that $\theta$ has already achieved the optimal statistical radius. Therefore, the following bounds hold with probability of at least $1 - \delta$,

$$\sup_{\theta \in \mathbb{B}(\theta^*, r)} \|\nabla \mathcal{L}_n(\theta)) - \nabla \mathcal{L}^0(\theta))\| \leq C_p r^{p-1} \sqrt{d \log(1/\delta)/n}, \tag{58}$$

$$\sup_{\theta \in \mathbb{B}(\theta^*, r)} \|\nabla^2 \mathcal{L}_n(\theta)) - \nabla^2 \mathcal{L}^0(\theta))\| \leq C_p r^{p-2} \sqrt{d \log(1/\delta)/n}. \tag{59}$$

In the following proof, we assume that all the results hold with probability of at least $1 - \delta$. Given the fact that the difference between the gradient and Hessian of the finite sample loss, denoted as $\mathcal{L}_n$, and the population loss with the optimal value at zero, denoted by $\mathcal{L}^0$, is of the order of statistical accuracy, and considering that the iterates generated by running BFGS on $\mathcal{L}^0$ converge to the optimal solution at a linear rate, we show that the iterates generated by running BFGS on the finite sample loss reach the statistical accuracy in a logarithmic number of iterations of the required statistical accuracy.

To simplify our presentation, we use $\{\theta_t\}_{t \geq 0}$ as the iterates generated by running BFGS on the population loss $\mathcal{L}^0$ and $\{\theta_t^n\}_{t \geq 0}$ as the iterates generated by running BFGS on the finite sample loss $\mathcal{L}_n$. We assume that $\theta_0 = \theta_0^n$, i.e., the first iterates are the same for both population loss and empirical loss. Similarly, we denote the inverse Hessian approximation matrix of BFGS applied to the loss $\mathcal{L}^0$ by $\{H_t\}_{t \geq 0}$ and the inverse Hessian approximation matrix of BFGS applied to the loss $\mathcal{L}_n$ by $\{H_t^n\}_{t \geq 0}$. We have that $H_0 = \nabla^2 \mathcal{L}^0(\theta_0)^{-1}$ and $H_0^n = \nabla^2 \mathcal{L}_n(\theta_0^n)^{-1}$. Given the results in Theorems 1 and 2, we know that the iterates generated by BFGS on $\mathcal{L}^0$ converge to the optimal solution at a linear rate, i.e.,

$$\theta_{t+1} - \theta^* = r_t(\theta_t - \theta^*), \qquad 0 < r_l \leq r_t \leq r_h < 1, \qquad \forall t \geq 0, \tag{60}$$

where $\{r_t\}_{t=0}^\infty$ are the linear convergence rates and $r_l, r_h \in (0, 1)$ are the lower and upper bounds of the corresponding linear convergence rates.

More precisely, we show that if the total number of iterations $T$ that we run BFGS on $\mathcal{L}_n$ is order of log of the inverse of statistical accuracy at most, i.e., $T = O(\log(n/d))$, then for any $t \leq T$ we can show that the difference between the iterates generated by running BFGS on $\mathcal{L}^0$ and $\mathcal{L}_n$ is controlled and bounded above by

$$\|\theta_t^n - \theta_t\| \leq c_t \|\theta_{t-1} - \theta^*\|^{1-p} \sqrt{d \log(1/\delta)/n}, \tag{61}$$

where $c_t = \Theta(\exp(t))$. Now given the fact that $\|\theta_{t-1} - \theta^*\|$ approaches zero at a linear rate, we will use induction to prove the main claim of Theorem 4. We assume that for any $t < T$, we have that

$$c_t \|\theta_t - \theta^*\|^{-p} \sqrt{d \log(1/\delta)/n} \leq \frac{1}{C_p} \leq 1. \tag{62}$$

Otherwise, our results in Theorem 4 simply follow. To prove the claim in (61) we will use an induction argument. Before doing that, in the upcoming sections we prove the following intermediate results that will be used in the induction argument.

**Lemma 3.** *We denote $\lambda_{\max}(A)$ and $\lambda_{\min}(A)$ as the largest and smallest eigenvalue of the matrix A, we have*

$$\|\nabla \mathcal{L}^0(\theta)\| \leq C_{p0} \|\theta - \theta^*\|^{2p-1}, \qquad \lambda_{\max}(\nabla^2 \mathcal{L}^0(\theta))) \leq C_{p1} \|\theta - \theta^*\|^{2p-2}, \qquad \forall \theta \in \mathbb{R}^d,$$
$$\|\nabla \mathcal{L}^0(\theta) - \nabla \mathcal{L}^0(\theta')\| \leq C_{p1} \|\theta - \theta'\|(\|\theta - \theta^*\| + \|\theta' - \theta^*\|)^{2p-2}, \qquad \forall \theta, \theta' \in \mathbb{R}^d,$$
$$\lambda_{\min}(\nabla^2 \mathcal{L}^0(\theta)) \geq C_{p2} \|\theta - \theta^*\|^{2p-2}, \qquad \lambda_{\min}(\nabla^2 \mathcal{L}_n(\theta_0)) \geq C_{p3} \|\theta_0 - \theta^*\|^{2p-2},$$

*where $C_{p0}, C_{p1}, C_{p2}$ and $C_{p3}$ are constants that only depend on $p$ and the last inequalities only holds for $\theta_0$.*

*Proof.* Recall that from (51) and (52), we have

$$\nabla \mathcal{L}^0(\theta) = 2p \mathbb{E}[(X^\top \theta)^{2p-1} X], \quad \nabla^2 \mathcal{L}^0(\theta) = 2p(2p-1) \mathbb{E}[(X^\top \theta)^{2p-2} X X^\top].$$

Using Cauchy–Schwarz inequality, we get that

$$\|\nabla\mathcal{L}^0(\theta))\| = 2p\|\mathbb{E}[(X^\top\theta)^{2p-1}X]\| \le 2p\|\mathbb{E}[(\|X\|\|\theta\|)^{2p-1}X]\| \le 2p\mathbb{E}[\|X\|^{2p-1}\|X\|]\|\theta\|^{2p-1}$$
$$\le 2p\mathbb{E}[\|X\|^{2p}](\|\theta-\theta^*\| + \|\theta^*\|)^{2p-1} \le 2p\mathbb{E}[\|X\|^{2p}](2\|\theta-\theta^*\|)^{2p-1} \le C_{p0}\|\theta-\theta^*\|^{2p-1},$$

where we use the fact that $\|\theta^*\| \le \|\theta-\theta^*\|$ for any $\theta$. If this does not hold, then we have $\|\theta-\theta^*\| < \|\theta^*\| \le C_1(\frac{d}{n})^{\frac{1}{2p}}$ by the definition of the low SNR regime, which indicates that $\theta$ has already achieved the optimal statistical radius. Similarly, we have that

$$\lambda_{\max}(\nabla^2\mathcal{L}^0(\theta))) = \|\nabla^2\mathcal{L}^0(\theta)\| = 2p(2p-1)\|\mathbb{E}[(X^\top\theta)^{2p-2}XX^\top]\| \le 2p(2p-1)\|\mathbb{E}[(\|X\|\|\theta\|)^{2p-2}XX^\top]\|$$
$$\le 2p(2p-1)\mathbb{E}[\|X\|^{2p-2}\|X\|^2]\|\theta\|^{2p-2} \le 2p(2p-1)\mathbb{E}[\|X\|^{2p}](\|\theta^*\| + \|\theta-\theta^*\|)^{2p-2} \le C_{p1}\|\theta-\theta^*\|^{2p-2},$$

where we use the same argument that $\|\theta^*\| \le \|\theta-\theta^*\|$. Using Taylor's Theorem, we have that

$$\nabla\mathcal{L}^0(\theta) - \nabla\mathcal{L}^0(\theta') = \nabla^2\mathcal{L}^0(\tau\theta + (1-\tau)\theta')(\theta-\theta'),$$

where $\tau \in [0,1]$. Therefore, for any $\theta, \theta'$, we have that

$$\|\nabla\mathcal{L}^0(\theta) - \nabla\mathcal{L}^0(\theta')\| \le \|\nabla^2\mathcal{L}^0(\tau\theta + (1-\tau)\theta')\|\|\theta-\theta'\| \le C_{p1}\|\tau\theta + (1-\tau)\theta' - \theta^*\|^{2p-2}\|\theta-\theta'\|$$
$$= C_{p1}\|\tau(\theta-\theta^*) + (1-\tau)(\theta'-\theta^*)\|^{2p-2}\|\theta-\theta'\| \le C_{p1}\|\theta-\theta'\|(\|\theta-\theta^*\| + \|\theta-\theta^*\|)^{2p-2}.$$

Notice that $\nabla^2\mathcal{L}^0(\theta)$ are positive definite for any $\theta$ and $\nabla^2\mathcal{L}_n(\theta_0)$ is positive definite, hence there exists $C_{p2}$ and $C_{p3}$ such that $\lambda_{\min}(\nabla^2\mathcal{L}^0(\theta)) \ge C_{p2}\|\theta-\theta^*\|^{2p-2}$ for any $\theta$ and $\lambda_{\min}(\nabla^2\mathcal{L}_n(\theta_0)) \ge C_{p3}\|\theta_0-\theta^*\|^{2p-2}$.

$\square$

### A.5.1  Induction base

Now we use the induction argument to prove the claim in (61). Since the initial iterate for running BFGS on $\mathcal{L}^0$ and $\mathcal{L}_n$ are the same, we have $\theta_0^n = \theta_0$. Now for the iterates generated after the first step of BFGS, we can show that the error between the iterates of two losses $\|\theta_1^n - \theta_1\|$ is bounded above by

$$\|\theta_1^n - \theta_1\| = \|(\nabla^2\mathcal{L}_n(\theta_0))^{-1}\nabla\mathcal{L}_n(\theta_0) - (\nabla^2\mathcal{L}^0(\theta_0))^{-1}\nabla\mathcal{L}^0(\theta_0)\|$$
$$\le\|\nabla^2\mathcal{L}_n(\theta_0)^{-1} - \nabla^2\mathcal{L}^0(\theta_0)^{-1}\|\|\nabla\mathcal{L}_n(\theta_0)\| + \|\nabla^2\mathcal{L}^0(\theta_0)^{-1}\|\|\nabla\mathcal{L}_n(\theta_0) - \nabla\mathcal{L}^0(\theta_0)\| \tag{63}$$

We observe that for invertible matrices $A$ and $B$, we have $(A^{-1} - B^{-1}) = A^{-1}(B - A)B^{-1}$. Given this observation and the results in Lemma 3 and (59), we can show that

$$\|\nabla^2\mathcal{L}_n(\theta_0)^{-1} - \nabla^2\mathcal{L}^0(\theta_0)^{-1}\| \le \|\nabla^2\mathcal{L}_n(\theta_0)^{-1}\|\|\nabla^2\mathcal{L}_n(\theta_0) - \nabla^2\mathcal{L}^0(\theta_0)\|\|\nabla^2\mathcal{L}^0(\theta_0)^{-1}\|$$
$$\le C_p\|\theta_0 - \theta^*\|^{2-3p}\sqrt{d\log(1/\delta)/n},$$

Applying this bound into (63) and using the results in Lemma 3, the bounds in (58), we obtain that

$$\|\theta_1^n - \theta_1\| \le C_p\|\theta_0 - \theta^*\|^{1-p}\sqrt{d\log(1/\delta)/n}. \tag{64}$$

Notice that from (64), we know that (61) holds for $t = 1$. Hence, the base of induction is complete.

### A.5.2  Induction hypothesis and step

Now we assume for any $k \le t$, (61) holds, i.e., we have $\|\theta_k^n - \theta_k\| \le c_k\|\theta_{k-1} - \theta^*\|^{1-p}\sqrt{d\log(1/\delta)/n}$. Consider $k = t + 1$, we have that

$$\|\theta_{t+1}^n - \theta_{t+1}\| = \|\theta_t^n - H_t^n\nabla\mathcal{L}_n(\theta_t^n) - \theta_t + H_t\nabla\mathcal{L}^0(\theta_t)\| \le \|\theta_t^n - \theta_t\| + \|H_t^n\nabla\mathcal{L}_n(\theta_t^n) - H_t\nabla\mathcal{L}^0(\theta_t)\|. \tag{65}$$

Recall update rules for the Hessian approximation matrices:

$$H_t^n = \left(I - \frac{s_{t-1}^n(u_{t-1}^n)^\top}{(u_{t-1}^n)^\top s_{t-1}^n}\right)H_{t-1}^n\left(I - \frac{u_{t-1}^n(s_{t-1}^n)^\top}{(s_{t-1}^n)^\top u_{t-1}^n}\right) + \frac{s_{t-1}^n(s_{t-1}^n)^\top}{(s_{t-1}^n)^\top u_{t-1}^n},$$

$$H_t = \left(I - \frac{s_{t-1}(u_{t-1})^\top}{(u_{t-1})^\top s_{t-1}}\right) H_{t-1} \left(I - \frac{u_{t-1}(s_{t-1})^\top}{(s_{t-1})^\top u_{t-1}}\right) + \frac{s_{t-1}(s_{t-1})^\top}{(s_{t-1})^\top u_{t-1}},$$

$$s_{t-1}^n = \theta_t^n - \theta_{t-1}^n, \quad u_{t-1}^n = \nabla\mathcal{L}_n(\theta_t^n) - \nabla\mathcal{L}_n(\theta_{t-1}^n) \quad s_{t-1} = \theta_t - \theta_{t-1}, \quad u_{t-1} = \nabla\mathcal{L}^0(\theta_t) - \nabla\mathcal{L}^0(\theta_{t-1}).$$

Moreover, based on (31) we have

$$\left(I - \frac{u_{t-1}s_{t-1}^\top}{s_{t-1}^\top u_{t-1}}\right)\nabla\mathcal{L}^0(\theta_t) = 0, \tag{66}$$

which implies that $H_t\nabla\mathcal{L}^0(\theta_t)$ can be simplified as $\frac{s_{t-1}s_{t-1}^\top}{s_{t-1}^\top u_{t-1}}\nabla\mathcal{L}^0(\theta_t)$. Now we proceed to bound the gap between the BFGS descent direction on $\mathcal{L}_n$ and $\mathcal{L}^0$, which can be upper bounded by

$$\|H_t^n\nabla\mathcal{L}_n(\theta_t^n) - H_t\nabla\mathcal{L}^0(\theta_t)\| = \left\|H_t^n\nabla\mathcal{L}_n(\theta_t^n) - \frac{s_{t-1}s_{t-1}^\top}{s_{t-1}^\top u_{t-1}}\nabla\mathcal{L}^0(\theta_t)\right\|$$

$$\leq \left\|\left(I - \frac{s_{t-1}^n(u_{t-1}^n)^\top}{(u_{t-1}^n)^\top s_{t-1}^n}\right)H_{t-1}^n\left(I - \frac{u_{t-1}^n(s_{t-1}^n)^\top}{(s_{t-1}^n)^\top u_{t-1}^n}\right)\nabla\mathcal{L}_n(\theta_t^n)\right\| + \left\|\frac{s_{t-1}^n(s_{t-1}^n)^\top}{(s_{t-1}^n)^\top u_{t-1}^n}\nabla\mathcal{L}_n(\theta_t^n) - \frac{s_{t-1}s_{t-1}^\top}{s_{t-1}^\top u_{t-1}}\nabla\mathcal{L}^0(\theta_t)\right\|, \tag{67}$$

Now we bound these two terms separately. The first term in (67) can be bounded above by

$$\left\|\left(I - \frac{s_{t-1}^n(u_{t-1}^n)^\top}{(u_{t-1}^n)^\top s_{t-1}^n}\right)H_{t-1}^n\left(I - \frac{u_{t-1}^n(s_{t-1}^n)^\top}{(s_{t-1}^n)^\top u_{t-1}^n}\right)\nabla\mathcal{L}_n(\theta_t^n)\right\|$$

$$\leq \left\|I - \frac{s_{t-1}^n(u_{t-1}^n)^\top}{(u_{t-1}^n)^\top s_{t-1}^n}\right\|\|H_{t-1}^n\|\left\|\left(I - \frac{u_{t-1}^n(s_{t-1}^n)^\top}{(s_{t-1}^n)^\top u_{t-1}^n}\right)\nabla\mathcal{L}_n(\theta_t^n)\right\| \tag{68}$$

$$\leq \|H_{t-1}^n\|\left\|\left(I - \frac{u_{t-1}^n(s_{t-1}^n)^\top}{(s_{t-1}^n)^\top u_{t-1}^n}\right)\nabla\mathcal{L}_n(\theta_t^n)\right\|,$$

where we use the fact that $\left\|I - \frac{s_{t-1}^n(u_{t-1}^n)^\top}{(u_{t-1}^n)^\top s_{t-1}^n}\right\| \leq 1$. Now using the fact that $\left(I - \frac{u_{t-1}s_{t-1}^\top}{s_{t-1}^\top u_{t-1}}\right)\nabla\mathcal{L}^0(\theta_t) = 0$, we can further upper bound $\left\|\left(I - \frac{u_{t-1}^n(s_{t-1}^n)^\top}{(s_{t-1}^n)^\top u_{t-1}^n}\right)\nabla\mathcal{L}_n(\theta_t^n)\right\|$ by

$$\left\|\left(I - \frac{u_{t-1}^n(s_{t-1}^n)^\top}{(s_{t-1}^n)^\top u_{t-1}^n}\right)\nabla\mathcal{L}_n(\theta_t^n)\right\| = \left\|\left(I - \frac{u_{t-1}^n(s_{t-1}^n)^\top}{(s_{t-1}^n)^\top u_{t-1}^n}\right)\nabla\mathcal{L}_n(\theta_t^n) - \left(I - \frac{u_{t-1}s_{t-1}^\top}{s_{t-1}^\top u_{t-1}}\right)\nabla\mathcal{L}^0(\theta_t)\right\|$$

$$\leq \|\nabla\mathcal{L}_n(\theta_t^n) - \nabla\mathcal{L}^0(\theta_t)\| + \left\|\frac{u_{t-1}^n s_{t-1}^n{}^\top\nabla\mathcal{L}_n(\theta_t^n)}{(s_{t-1}^n)^\top u_{t-1}^n} - \frac{u_{t-1}^n s_{t-1}^\top\nabla\mathcal{L}^0(\theta_t)}{s_{t-1}^\top u_{t-1}}\right\| \tag{69}$$

$$+ \left\|\frac{u_{t-1}^n s_{t-1}^\top\nabla\mathcal{L}^0(\theta_t)}{s_{t-1}^\top u_{t-1}} - \frac{u_{t-1}s_{t-1}^\top\nabla\mathcal{L}^0(\theta_t)}{s_{t-1}^\top u_{t-1}}\right\|$$

$$\leq \|\nabla\mathcal{L}_n(\theta_t^n) - \nabla\mathcal{L}^0(\theta_t)\| + \|u_{t-1}^n\|\left|\frac{(s_{t-1}^n)^\top\nabla\mathcal{L}_n(\theta_t^n)}{(s_{t-1}^n)^\top u_{t-1}^n} - \frac{s_{t-1}^\top\nabla\mathcal{L}^0(\theta_t)}{s_{t-1}^\top u_{t-1}}\right| + \|u_{t-1}^n - u_{t-1}\|\left|\frac{s_{t-1}^\top\nabla\mathcal{L}^0(\theta_t)}{s_{t-1}^\top u_{t-1}}\right|,$$

Next we proceed to bound the second term in (67) . The second term can be bounded as

$$\left\|\frac{s_{t-1}^n(s_{t-1}^n)^\top}{(s_{t-1}^n)^\top u_{t-1}^n}\nabla\mathcal{L}_n(\theta_t^n) - \frac{s_{t-1}s_{t-1}^\top}{s_{t-1}^\top u_{t-1}}\nabla\mathcal{L}^0(\theta_t)\right\|$$

$$\leq \left\|\frac{s_{t-1}^n(s_{t-1}^n)^\top}{(s_{t-1}^n)^\top u_{t-1}^n}\nabla\mathcal{L}_n(\theta_t^n) - \frac{s_{t-1}^n s_{t-1}^\top}{s_{t-1}^\top u_{t-1}}\nabla\mathcal{L}^0(\theta_t)\right\| + \left\|\frac{s_{t-1}^n s_{t-1}^\top}{s_{t-1}^\top u_{t-1}}\nabla\mathcal{L}^0(\theta_t) - \frac{s_{t-1}s_{t-1}^\top}{s_{t-1}^\top u_{t-1}}\nabla\mathcal{L}^0(\theta_t)\right\|$$

$$\leq \|s_{t-1}^n\|\left|\frac{(s_{t-1}^n)^\top\nabla\mathcal{L}_n(\theta_t^n)}{(s_{t-1}^n)^\top u_{t-1}^n} - \frac{s_{t-1}^\top\nabla\mathcal{L}^0(\theta_t)}{s_{t-1}^\top u_{t-1}}\right| + \|s_{t-1} - s_{t-1}^n\|\left|\frac{s_{t-1}^\top\nabla\mathcal{L}^0(\theta_t)}{s_{t-1}^\top u_{t-1}}\right|. \tag{70}$$

Putting together the upper bounds in (68), (69), and (70) into (67), we obtain that

$$
\begin{aligned}
&\|H_t^n \nabla \mathcal{L}_n(\theta_t^n) - H_t \nabla \mathcal{L}^0(\theta_t)\| \\
&\leq \|H_{t-1}^n\|\|\nabla \mathcal{L}_n(\theta_t^n) - \nabla \mathcal{L}^0(\theta_t)\| + (\|H_{t-1}^n\|\|u_{t-1}^n\| + \|s_{t-1}^n\|) \left| \frac{(s_{t-1}^n)^\top \nabla \mathcal{L}_n(\theta_t^n)}{(s_{t-1}^n)^\top u_{t-1}^n} - \frac{s_{t-1}^\top \nabla \mathcal{L}^0(\theta_t)}{s_{t-1}^\top u_{t-1}} \right| \\
&\quad + (\|H_{t-1}^n\|\|u_{t-1}^n - u_{t-1}\| + \|s_{t-1} - s_{t-1}^n\|) \frac{s_{t-1}^\top \nabla \mathcal{L}^0(\theta_t)}{s_{t-1}^\top u_{t-1}}|.
\end{aligned}
\tag{71}
$$

In the following lemmas, we establish upper bounds for the expressions in the right hand side of (71).

**Lemma 4.** *The norm of variable variation for the finite sample loss $s_{t-1}^n$ and its difference form the variable variation for the population loss $s_{t-1}$ are bounded above by*

$$
\|s_{t-1}^n - s_{t-1}\| \leq (c_t + c_{t-1})\|\theta_{t-1} - \theta^*\|^{1-p}\sqrt{d\log(1/\delta)/n}.
$$

*Moreover, this result implies that*

$$
\|s_{t-1}^n\| \leq \|\theta_{t-1} - \theta^*\|.
$$

*Proof.* The first result simply follows from the fact that

$$
\begin{aligned}
\|s_{t-1}^n - s_{t-1}\| &\leq \|\theta_t^n - \theta_t\| + \|\theta_{t-1}^n - \theta_{t-1}\| \\
&\leq (c_t\|\theta_{t-1} - \theta^*\|^{1-p} + c_{t-1}\|\theta_{t-2} - \theta^*\|^{1-p})\sqrt{d\log(1/\delta)/n} \leq (c_t + c_{t-1}r_{t-1}^{p-1})\|\theta_{t-1} - \theta^*\|^{1-p}\sqrt{d\log(1/\delta)/n} \\
&\leq (c_t + c_{t-1})\|\theta_{t-1} - \theta^*\|^{1-p}\sqrt{d\log(1/\delta)/n},
\end{aligned}
$$

where we used the induction assumption (61) and the linear convergence results (60) for the iterates generated on the population loss. The second claim can be also proved following

$$
\begin{aligned}
\|s_{t-1}^n\| &\leq \|s_{t-1}\| + \|s_{t-1}^n - s_{t-1}\| = \|\theta_t - \theta^* - (\theta_{t-1} - \theta^*)\| + \|\theta_t^n - \theta_t\| + \|\theta_{t-1}^n - \theta_{t-1}\| \\
&\leq (1 - r_{t-1})\|\theta_{t-1} - \theta^*\| + c_t\|\theta_{t-1} - \theta^*\|^{1-p}\sqrt{d\log(1/\delta)/n} + c_{t-1}\|\theta_{t-2} - \theta^*\|^{1-p}\sqrt{d\log(1/\delta)/n} \\
&\leq (1 - r_{t-1})\|\theta_{t-1} - \theta^*\| + \frac{1 + 1/r_{t-2}}{C_p}\|\theta_{t-1} - \theta^*\| \leq \|\theta_{t-1} - \theta^*\|,
\end{aligned}
$$

where we used the induction assumption (61), linear convergence results (60) and the assumption (62) that $c_t\|\theta_{t-1} - \theta^*\|^{-p}\sqrt{d\log(1/\delta)/n} \leq \frac{1}{C_p}$ with sufficiently large $C_p$ to make the last inequality hold. $\qquad\square$

**Lemma 5.** *The gap between the population loss and its finite sample version evaluated at the current iterates are bounded above by*

$$
\|\nabla \mathcal{L}_n(\theta_t^n) - \nabla \mathcal{L}^0(\theta_t)\| \leq (C_p + C_p c_t)\|\theta_{t-1} - \theta^*\|^{p-1}\sqrt{d\log(1/\delta)/n},
$$

*Moreover, this result implies that*

$$
\|\nabla \mathcal{L}_n(\theta_t^n)\| \leq C_p\|\theta_t - \theta^*\|^{2p-1}.
$$

*Proof.* We have that

$$
\begin{aligned}
\|\nabla \mathcal{L}_n(\theta_t^n) - \nabla \mathcal{L}^0(\theta_t^n)\| &\leq C_p\|\theta_t^n - \theta^*\|^{p-1}\sqrt{d\log(1/\delta)/n} \leq C_p(\|\theta_t^n - \theta_t\| + \|\theta_t - \theta^*\|)^{p-1}\sqrt{d\log(1/\delta)/n} \\
&\leq C_p(c_t\|\theta_{t-1} - \theta^*\|^{1-p}\sqrt{d\log(1/\delta)/n} + r_{t-1}\|\theta_{t-1} - \theta^*\|)^{p-1}\sqrt{d\log(1/\delta)/n} \\
&\leq C_p(\|\theta_{t-1} - \theta^*\| + \|\theta_{t-1} - \theta^*\|)^{p-1}\sqrt{d\log(1/\delta)/n} \leq C_p(\|\theta_{t-1} - \theta^*\|)^{p-1}\sqrt{d\log(1/\delta)/n},
\end{aligned}
$$

where the first inequality is due to (58), the third inequality is due to the induction hypothesis (61) and linear convergence results in (60) and the forth inequality is due to assumption (62) and $r_{t-1} \leq 1$. We also have

$$\|\nabla \mathcal{L}^0(\theta_t^n) - \nabla \mathcal{L}^0(\theta_t)\| \leq C_p \|\theta_t^n - \theta_t\| (\|\theta_t^n - \theta^*\| + \|\theta_t - \theta^*\|)^{2p-2}$$
$$\leq C_p c_t \|\theta_{t-1} - \theta^*\|^{1-p} \sqrt{d \log(1/\delta)/n} (\|\theta_t^n - \theta_t\| + 2\|\theta_t - \theta^*\|)^{2p-2}$$
$$\leq C_p c_t \|\theta_{t-1} - \theta^*\|^{1-p} \sqrt{d \log(1/\delta)/n} (c_t \|\theta_{t-1} - \theta^*\|^{1-p} \sqrt{d \log(1/\delta)/n} + 2r_{t-1} \|\theta_{t-1} - \theta^*\|)^{2p-2}$$
$$\leq C_p c_t \|\theta_{t-1} - \theta^*\|^{1-p} \sqrt{d \log(1/\delta)/n} (\|\theta_{t-1} - \theta^*\| + 2\|\theta_{t-1} - \theta^*\|)^{2p-2}$$
$$\leq C_p c_t \|\theta_{t-1} - \theta^*\|^{p-1} \sqrt{d \log(1/\delta)/n},$$

where the first inequality is due to results in Lemma 3, the second inequality is due to the induction hypothesis (61) and $\|\theta_t^n - \theta^*\| \leq \|\theta_t^n - \theta_t\| + \|\theta_t - \theta^*\|$, the third inequality is due to the induction hypothesis (61) and the linear convergence results (60) and the forth inequality is due to the assumption (62) and $r_{t-1} \leq 1$. The first claim follows using the fact that

$$\|\nabla \mathcal{L}_n(\theta_t^n) - \nabla \mathcal{L}^0(\theta_t)\| \leq \|\nabla \mathcal{L}_n(\theta_t^n) - \nabla \mathcal{L}^0(\theta_t^n)\| + \|\nabla \mathcal{L}^0(\theta_t^n) - \nabla \mathcal{L}^0(\theta_t)\|$$
$$\leq (C_p + C_p c_t) \|\theta_{t-1} - \theta^*\|^{p-1} \sqrt{d \log(1/\delta)/n}.$$

Given this result, the second claim simply follows from the fact that

$$\|\nabla \mathcal{L}_n(\theta_t^n)\| \leq \|\nabla \mathcal{L}^0(\theta_t)\| + \|\nabla \mathcal{L}_n(\theta_t^n) - \nabla \mathcal{L}^0(\theta_t)\|$$
$$\leq C_p \|\theta_t - \theta^*\|^{2p-1} + (C_p + C_p c_t) \|\theta_{t-1} - \theta^*\|^{p-1} \sqrt{d \log(1/\delta)/n} \leq C_p \|\theta_t - \theta^*\|^{2p-1} + C_p \|\theta_{t-1} - \theta^*\|^{2p-1}$$
$$\leq C_p \|\theta_t - \theta^*\|^{2p-1} + \frac{C_p}{r_{t-1}^{2p-1}} \|\theta_t - \theta^*\|^{2p-1} \leq C_p \|\theta_t - \theta^*\|^{2p-1},$$

where we used results from Lemma 3, the linear convergence rate in (60) and the assumption (62) that $c_t \|\theta_{t-1} - \theta^*\|^{-p} \sqrt{d \log(1/\delta)/n} \leq \frac{1}{C_p} \leq 1$ again. $\qquad \square$

**Lemma 6.** *The norm of gradient variation for the finite sample loss $u_{t-1}^n$ and its difference form the gradient variation for the population loss $u_{t-1}$ are bounded above by*

$$\|u_{t-1}^n - u_{t-1}\| \leq (C_p + C_p c_t + C_p c_{t-1}) \|\theta_{t-1} - \theta^*\|^{p-1} \sqrt{d \log(1/\delta)/n}.$$

*Moreover, this result implies that*

$$\|u_{t-1}^n\| \leq C_p \|\theta_{t-1} - \theta^*\|^{2p-1}.$$

*Proof.* The first claim simply follows from the following bounds,

$$\|u_{t-1}^n - u_{t-1}\| \leq \|\nabla \mathcal{L}_n(\theta_t^n) - \nabla \mathcal{L}^0(\theta_t)\| + \|\nabla \mathcal{L}_n(\theta_{t-1}^n) - \nabla \mathcal{L}^0(\theta_{t-1})\|$$
$$\leq (C_p + C_p c_t) \|\theta_{t-1} - \theta^*\|^{p-1} \sqrt{d \log(1/\delta)/n} + (C_p + C_p c_{t-1}) \|\theta_{t-2} - \theta^*\|^{p-1} \sqrt{d \log(1/\delta)/n}$$
$$\leq (C_p + C_p c_t) \|\theta_{t-1} - \theta^*\|^{p-1} \sqrt{d \log(1/\delta)/n} + \frac{C_p + C_p c_{t-1}}{r_{t-2}^{p-1}} \|\theta_{t-1} - \theta^*\|^{p-1} \sqrt{d \log(1/\delta)/n}$$
$$\leq (C_p + C_p c_t + C_p c_{t-1}) \|\theta_{t-1} - \theta^*\|^{p-1} \sqrt{d \log(1/\delta)/n}.$$

where we used the results in Lemma 5 and the linear convergence results (60) of BFGS on the population loss. The second claim simply follows from,

$$\|u_{t-1}^n\| \le \|u_{t-1}^n - u_{t-1}\| + \|u_{t-1}\|$$

$$\le \|\nabla \mathcal{L}_n(\theta_t^n) - \nabla \mathcal{L}^0(\theta_t)\| + \|\nabla \mathcal{L}_n(\theta_{t-1}^n) - \nabla \mathcal{L}^0(\theta_{t-1})\| + \|\nabla \mathcal{L}^0(\theta_t) - \nabla \mathcal{L}^0(\theta_{t-1})\|$$

$$\le (C_p + C_p c_t)\|\theta_{t-1} - \theta^*\|^{p-1}\sqrt{d\log(1/\delta)/n} + (C_p + C_p c_{t-1})\|\theta_{t-2} - \theta^*\|^{p-1}\sqrt{d\log(1/\delta)/n}$$

$$+ C_p\|\theta_t - \theta_{t-1}\|(\|\theta_t - \theta^*\| + \|\theta_{t-1} - \theta^*\|)^{2p-2}$$

$$\le C_p\|\theta_{t-1} - \theta^*\|^{2p-1} + C_p\|\theta_{t-2} - \theta^*\|^{2p-1} + C_p\|\theta_t - \theta^* - (\theta_{t-1} - \theta^*)\|(\|\theta_t - \theta^*\| + \|\theta_{t-1} - \theta^*\|)^{2p-2}$$

$$\le C_p\|\theta_{t-1} - \theta^*\|^{2p-1} + \frac{C_p}{r_{t-2}^{2p-1}}\|\theta_{t-1} - \theta^*\|^{2p-1}$$

$$+ C_p\|(1 - r_{t-1})(\theta_{t-1} - \theta^*)\|(r_{t-1}\|\theta_{t-1} - \theta^*\| + \|\theta_{t-1} - \theta^*\|)^{2p-2}$$

$$= C_p\|\theta_{t-1} - \theta^*\|^{2p-1} + \frac{C_p}{r_{t-2}^{2p-1}}\|\theta_{t-1} - \theta^*\|^{2p-1} + C_p(1 - r_{t-1})(1 + r_{t-1})^{2p-2}\|\theta_{t-1} - \theta^*\|^{2p-1}$$

$$\le C_p\|\theta_{t-1} - \theta^*\|^{2p-1},$$

where the third inequality is due to results in Lemma 5 and Lemma 3, the forth inequality is due to assumption (62) and the fifth inequality is due to linear convergence results (60). $\square$

**Lemma 7.** *We have the following bounds:*

$$|(s_{t-1}^n)^\top \nabla \mathcal{L}_n(\theta_t^n) - s_{t-1}^\top \nabla \mathcal{L}^0(\theta_t)| \le (C_p + C_p c_t + C_p c_{t-1})\|\theta_{t-1} - \theta^*\|^p \sqrt{d\log(1/\delta)/n}.$$

$$|(s_{t-1}^n)^\top (u_{t-1}^n) - s_{t-1}^\top u_{t-1}| \le (C_p + C_p c_t + C_p c_{t-1})\|\theta_{t-1} - \theta^*\|^p \sqrt{d\log(1/\delta)/n}.$$

*Proof.* The first claim holds since

$$|(s_{t-1}^n)^\top \nabla \mathcal{L}_n(\theta_t^n) - s_{t-1}^\top \nabla \mathcal{L}^0(\theta_t)| \le \|s_{t-1}^n - s_{t-1}\|\|\nabla \mathcal{L}_n(\theta_t^n)\| + \|s_{t-1}\|\|\nabla \mathcal{L}_n(\theta_t^n) - \nabla \mathcal{L}^0(\theta_t)\|$$

$$\le (c_t + c_{t-1})\|\theta_{t-1} - \theta^*\|^{1-p}\sqrt{d\log(1/\delta)/n}C_p\|\theta_t - \theta^*\|^{2p-1}$$

$$+ \|\theta_{t-1} - \theta^*\|(C_p + C_p c_t)\|\theta_{t-1} - \theta^*\|^{p-1}\sqrt{d\log(1/\delta)/n}$$

$$\le (C_p c_t + C_p c_{t-1})r_{t-1}^{2p-1}\|\theta_{t-1} - \theta^*\|^p\sqrt{d\log(1/\delta)/n} + (C_p + C_p c_t)\|\theta_{t-1} - \theta^*\|^p\sqrt{d\log(1/\delta)/n}$$

$$\le (C_p c_t + C_p c_{t-1})\|\theta_{t-1} - \theta^*\|^p\sqrt{d\log(1/\delta)/n} + (C_p + C_p c_t)\|\theta_{t-1} - \theta^*\|^p\sqrt{d\log(1/\delta)/n}$$

$$\le (C_p + C_p c_t + C_p c_{t-1})\|\theta_{t-1} - \theta^*\|^p\sqrt{d\log(1/\delta)/n},$$

where the second inequality is due to results in Lemma 4 and Lemma 5, the third inequality is due to linear convergence rates (60) and the forth inequality is due to $r_{t-1} \le 1$. The second claim holds since

$$|(s_{t-1}^n)^\top (u_{t-1}^n) - s_{t-1}^\top u_{t-1}| \le \|s_{t-1}^n - s_{t-1}\|\|u_{t-1}^n\| + \|s_{t-1}\|\|u_{t-1}^n - u_{t-1}\|$$

$$\le (c_t + c_{t-1})\|\theta_{t-1} - \theta^*\|^{1-p}\sqrt{d\log(1/\delta)/n}C_p\|\theta_{t-1} - \theta^*\|^{2p-1}$$

$$+ \|\theta_t - \theta_{t-1}\|(C_p + C_p c_t + C_p c_{t-1})\|\theta_{t-1} - \theta^*\|^{p-1}\sqrt{d\log(1/\delta)/n}$$

$$= (C_p c_t + C_p c_{t-1})\|\theta_{t-1} - \theta^*\|^p\sqrt{d\log(1/\delta)/n}$$

$$+ \|\theta_t - \theta^* - \theta_{t-1} + \theta^*\|(C_p + C_p c_t + C_p c_{t-1})\|\theta_{t-1} - \theta^*\|^{p-1}\sqrt{d\log(1/\delta)/n}$$

$$\le (C_p c_t + C_p c_{t-1})\|\theta_{t-1} - \theta^*\|^p\sqrt{d\log(1/\delta)/n}$$

$$+ (1 - r_{t-1})\|\theta_{t-1} - \theta^*\|(C_p + C_p c_t + C_p c_{t-1})\|\theta_{t-1} - \theta^*\|^{p-1}\sqrt{d\log(1/\delta)/n}$$

$$\le (C_p c_t + C_p c_{t-1})\|\theta_{t-1} - \theta^*\|^p\sqrt{d\log(1/\delta)/n} + (C_p + C_p c_t + C_p c_{t-1})\|\theta_{t-1} - \theta^*\|^p\sqrt{d\log(1/\delta)/n}$$

$$\le (C_p + C_p c_t + C_p c_{t-1})\|\theta_{t-1} - \theta^*\|^p\sqrt{d\log(1/\delta)/n},$$

where the second inequality is due to results in Lemma 4 and Lemma 6, the third inequality is due to linear convergence rates (60). $\square$

**Lemma 8.** *The following bounds hold:*

$$|s_{t-1}^\top \nabla \mathcal{L}^0(\theta_t)| \le C_p \|\theta_t - \theta^*\|^{2p}, \qquad |s_{t-1}^\top u_{t-1}| \ge C_p \|\theta_t - \theta^*\|^{2p}, \qquad |(s_{t-1}^n)^\top u_{t-1}^n| \ge C_p \|\theta_t - \theta^*\|^{2p}.$$

*Proof.* The first claim holds since

$$|s_{t-1}^\top \nabla \mathcal{L}^0(\theta_t)| = |(\theta_t - \theta_{t-1})^\top \nabla \mathcal{L}^0(\theta_t)| = |(\theta_t - \theta^* - \theta_{t-1} + \theta^*)^\top \nabla \mathcal{L}^0(\theta_t)| = |(1 - \frac{1}{r_{t-1}})(\theta_t - \theta^*)^\top \nabla \mathcal{L}^0(\theta_t)|$$

$$\le |1 - \frac{1}{r_{t-1}}| \|\theta_t - \theta^*\| \|\nabla \mathcal{L}^0(\theta_t)\| \le |1 - \frac{1}{r_h}| \|\theta_t - \theta^*\| C_p \|\theta_t - \theta^*\|^{2p-1} \le C_p \|\theta_t - \theta^*\|^{2p},$$

where we use the linear convergence results (60) and the results in Lemma 3. The second claim holds since

$$|s_{t-1}^\top u_{t-1}| = |(\theta_t - \theta_{t-1})^\top (\nabla \mathcal{L}^0(\theta_t) - \nabla \mathcal{L}^0(\theta_{t-1}))|$$
$$= |(\theta_t - \theta^* - \theta_{t-1} + \theta^*)^\top \nabla^2 \mathcal{L}^0(\tau \theta_t + (1 - \tau)\theta_{t-1})(\theta_t - \theta_{t-1})|$$
$$= |(1 - \frac{1}{r_{t-1}})(\theta_t - \theta^*)^\top \nabla^2 \mathcal{L}^0(\tau \theta_t + (1 - \tau)\theta_{t-1})(\theta_t - \theta^* - \theta_{t-1} + \theta^*)|$$
$$= |(1 - \frac{1}{r_{t-1}})^2 (\theta_t - \theta^*)^\top \nabla^2 \mathcal{L}^0(\tau \theta_t + (1 - \tau)\theta_{t-1})(\theta_t - \theta^*)|$$
$$\ge (1 - \frac{1}{r_{t-1}})^2 \|\theta_t - \theta^*\|^2 \lambda_{min}(\nabla^2 \mathcal{L}^0(\tau \theta_t + (1 - \tau)\theta_{t-1}))$$
$$\ge (1 - \frac{1}{r_l})^2 \|\theta_t - \theta^*\|^2 C_p \|\tau \theta_t + (1 - \tau)\theta_{t-1} - \theta^*\|^{2p-2} \ge C_p \|\theta_t - \theta^*\|^2 \|\tau(\theta_t - \theta^*) + (1 - \tau)(\theta_{t-1} - \theta^*)\|^{2p-2}$$
$$\ge C_p \|\theta_t - \theta^*\|^2 \|(\tau + \frac{1 - \tau}{r_{t-1}})(\theta_t - \theta^*)\|^{2p-2} \ge C_p(\tau + \frac{1 - \tau}{r_h})^{2p-2} \|\theta_t - \theta^*\|^{2p} \ge C_p \|\theta_t - \theta^*\|^{2p},$$

where $\tau \in [0, 1]$ and we use the Taylor's Theorem, the linear convergence rates (60) and results in Lemma 3. The last cliam holds since

$$|(s_{t-1}^n)^\top u_{t-1}^n| = |(s_{t-1}^n)^\top u_{t-1}^n - s_{t-1}^\top u_{t-1} + s_{t-1}^\top u_{t-1}| \ge |s_{t-1}^\top u_{t-1}| - |(s_{t-1}^n)^\top u_{t-1}^n - s_{t-1}^\top u_{t-1}|$$
$$\ge C_p \|\theta_t - \theta^*\|^{2p} - (C_p + C_p c_t + C_p c_{t-1}) \|\theta_{t-1} - \theta^*\|^p \sqrt{d \log(1/\delta)/n}$$
$$\ge C_p \|\theta_t - \theta^*\|^{2p} - (C_p + C_p c_t) \frac{1}{r_{t-1}^p} \|\theta_t - \theta^*\|^p \sqrt{d \log(1/\delta)/n} - C_p c_{t-1} \|\theta_{t-1} - \theta^*\|^p \sqrt{d \log(1/\delta)/n}$$
$$\ge C_p \|\theta_t - \theta^*\|^{2p} - \|\theta_t - \theta^*\|^{2p} - \|\theta_t - \theta^*\|^{2p} \ge C_p \|\theta_t - \theta^*\|^{2p},$$

where we use the second claim, the results from Lemma 7 and the assumption (62). $\qquad \square$

**Lemma 9.** *We have the following bounds:*

$$\left|\frac{s_{t-1}^\top \nabla \mathcal{L}^0(\theta_t)}{s_{t-1}^\top u_{t-1}}\right| \le C_p,$$

$$\left|\frac{(s_{t-1}^n)^\top \nabla \mathcal{L}_n(\theta_t^n)}{(s_{t-1}^n)^\top u_{t-1}^n} - \frac{s_{t-1}^\top \nabla \mathcal{L}^0(\theta_t)}{s_{t-1}^\top u_{t-1}}\right| \le (C_p + C_p c_t + C_p c_{t-1}) \|\theta_{t-1} - \theta^*\|^{-p} \sqrt{d \log(1/\delta)/n}.$$

*Proof.* The first claim holds since

$$\left|\frac{s_{t-1}^\top \nabla \mathcal{L}^0(\theta_t)}{s_{t-1}^\top u_{t-1}}\right| = \frac{|s_{t-1}^\top \nabla \mathcal{L}^0(\theta_t)|}{|s_{t-1}^\top u_{t-1}|} \le \frac{C_p \|\theta_{t-1} - \theta^*\|^{2p}}{C_p \|\theta_{t-1} - \theta^*\|^{2p}} \le C_p,$$

where we use the results in Lemma 8. The second claim holds since

$$\left| \frac{(s_{t-1}^n)^\top \nabla \mathcal{L}_n(\theta_t^n)}{(s_{t-1}^n)^\top u_{t-1}^n} - \frac{s_{t-1}^\top \nabla \mathcal{L}^0(\theta_t)}{s_{t-1}^\top u_{t-1}} \right|$$

$$\leq \frac{|(s_{t-1}^n)^\top \nabla \mathcal{L}_n(\theta_t^n) - s_{t-1}^\top \nabla \mathcal{L}^0(\theta_t)||s_{t-1}^\top u_{t-1}| + |s_{t-1}^\top \nabla \mathcal{L}^0(\theta_t)||s_{t-1}^\top u_{t-1} - (s_{t-1}^n)^\top u_{t-1}^n|}{|(s_{t-1}^n)^\top u_{t-1}^n s_{t-1}^\top u_{t-1}|}$$

$$\leq \frac{|(s_{t-1}^n)^\top \nabla \mathcal{L}_n(\theta_t^n) - s_{t-1}^\top \nabla \mathcal{L}^0(\theta_t)|}{|(s_{t-1}^n)^\top u_{t-1}^n|} + \frac{|s_{t-1}^\top \nabla \mathcal{L}^0(\theta_t)|}{|s_{t-1}^\top u_{t-1}|} \frac{|s_{t-1}^\top u_{t-1} - (s_{t-1}^n)^\top u_{t-1}^n|}{|(s_{t-1}^n)^\top u_{t-1}^n|}$$

$$\leq \frac{(C_p + C_p c_t + C_p c_{t-1})\|\theta_{t-1} - \theta^*\|^p \sqrt{d \log(1/\delta)/n}}{C_p \|\theta_t - \theta^*\|^{2p}} + C_p \frac{(C_p + C_p c_t + C_p c_{t-1})\|\theta_{t-1} - \theta^*\|^p \sqrt{d \log(1/\delta)/n}}{C_p \|\theta_{t-1} - \theta^*\|^{2p}}$$

$$\leq \frac{C_p + C_p c_t + C_p c_{t-1}}{r_{t-1}^{2p}} \|\theta_{t-1} - \theta^*\|^{-p} \sqrt{d \log(1/\delta)/n} + (C_p + C_p c_t + C_p c_{t-1})\|\theta_{t-1} - \theta^*\|^{-p} \sqrt{d \log(1/\delta)/n}$$

$$\leq (C_p + C_p c_t + C_p c_{t-1})\|\theta_{t-1} - \theta^*\|^{-p} \sqrt{d \log(1/\delta)/n},$$

where we use the results from Lemma 7 and Lemma 8. $\square$

Finally, we present an upper bound for the norm of the inverse Hessian approximation matrix $H_t$.

**Lemma 10.** *The norm of the inverse Hessian approximation matrix is upper bounded by*

$$\|H_{t-1}^n\| \leq C_p \|\theta_{t-1} - \theta^*\|^{2-2p}.$$

*Proof.* Recall the update of $H_t^n$,

$$H_t^n = \left( I - \frac{s_{t-1}^n (u_{t-1}^n)^\top}{(u_{t-1}^n)^\top s_{t-1}^n} \right) H_{t-1}^n \left( I - \frac{u_{t-1}^n (s_{t-1}^n)^\top}{(s_{t-1}^n)^\top u_{t-1}^n} \right) + \frac{s_{t-1}^n (s_{t-1}^n)^\top}{(s_{t-1}^n)^\top u_{t-1}^n}.$$

With the property of $\|I - \frac{s_{t-1}^n (u_{t-1}^n)^\top}{(u_{t-1}^n)^\top s_{t-1}^n}\| \leq 1$ and $\|I - \frac{u_{t-1}^n (s_{t-1}^n)^\top}{(s_{t-1}^n)^\top u_{t-1}^n}\| \leq 1$, we have that

$$\|H_t^n\| \leq \|I - \frac{s_{t-1}^n (u_{t-1}^n)^\top}{(u_{t-1}^n)^\top s_{t-1}^n}\| \|H_{t-1}^n\| \|I - \frac{u_{t-1}^n (s_{t-1}^n)^\top}{(s_{t-1}^n)^\top u_{t-1}^n}\| + \|\frac{s_{t-1}^n (s_{t-1}^n)^\top}{(s_{t-1}^n)^\top u_{t-1}^n}\|$$

$$\leq \|H_{t-1}^n\| + \frac{\|s_{t-1}^n\|^2}{(s_{t-1}^n)^\top u_{t-1}^n} \leq \left\| \left( \nabla^2 \mathcal{L}_n(\theta_0) \right)^{-1} \right\| + \sum_{i=0}^{t-1} \frac{\|s_i^n\|^2}{(s_i^n)^\top (u_i^n)}.$$

From results in Lemma 4 and Lemma 8, we know that for any $0 \leq i \leq t-1$,

$$\frac{\|s_i^n\|^2}{(s_i^n)^\top (u_i^n)} \leq \frac{\|\theta_i - \theta^*\|^2}{C_p \|\theta_i - \theta^*\|^{2p}} \leq C_p \|\theta_i - \theta^*\|^{2-2p}.$$

Hence, using linear convergence results in (60), we have that for all $0 \leq i \leq t-1$,

$$\|\theta_{t-1} - \theta^*\| = \prod_{j=i}^{t-2} r_j \|\theta_i - \theta^*\| \leq r_h^{t-1-i} \|\theta_i - \theta^*\|,$$

$$\|\theta_i - \theta^*\| \geq r_h^{i+1-t} \|\theta_{t-1} - \theta^*\|, \qquad \|\theta_i - \theta^*\|^{2-2p} \leq r_h^{(2p-2)(t-1-i)} \|\theta_{t-1} - \theta^*\|^{2-2p},$$

$$\sum_{i=0}^{t-1} \|\theta_i - \theta^*\|^{2-2p} \leq \sum_{i=0}^{t-1} r_h^{(2p-2)(t-1-i)} \|\theta_{t-1} - \theta^*\|^{2-2p} \leq \frac{1}{1 - r_h^{2p-2}} \|\theta_{t-1} - \theta^*\|^{2-2p}.$$

Therefore, we obtain that

$$\|H_t^n\| \le \left\|\left(\nabla^2 \mathcal{L}_n(\theta_0)\right)^{-1}\right\| + \sum_{i=0}^{t-1} \frac{\|s_i^n\|^2}{(s_i^n)^\top (u_i^n)} \le \frac{1}{\lambda_{\min}(\nabla^2 \mathcal{L}_n(\theta_0))} + \sum_{i=0}^{t-1} C_p \|\theta_i - \theta^*\|^{2-2p}$$

$$\le \frac{1}{C_p \|\theta_0 - \theta^*\|^{2p-2}} + C_p \frac{1}{1 - r_h^{2p-2}} \|\theta_{t-1} - \theta^*\|^{2-2p} \le \left(\frac{r_h^{(2p-2)(t-1)}}{C_p} + \frac{C_p}{1 - r_h^{2p-2}}\right) \|\theta_{t-1} - \theta^*\|^{2-2p}$$

$$\le C_p \|\theta_{t-1} - \theta^*\|^{2-2p}.$$

Hence, we have that

$$\|H_{t-1}^n\| \le C_p \|\theta_{t-2} - \theta^*\|^{2-2p} \le C_p \frac{1}{r_{t-2}^{2-2p}} \|\theta_{t-1} - \theta^*\|^{2-2p} \le C_p r_h^{2p-2} \|\theta_{t-1} - \theta^*\|^{2-2p} \le C_p \|\theta_{t-1} - \theta^*\|^{2-2p}.$$

$\square$

With all the above lemmas, we can complete the induction hypothesis. Applying results in Lemma 4, 5, 6, 9, and 10 into (71), we have that

$$\|H_t^n \nabla \mathcal{L}_n(\theta_t^n) - H_t \nabla \mathcal{L}^0(\theta_t)\|$$

$$\le \|H_{t-1}^n\| \|\nabla \mathcal{L}_n(\theta_t^n) - \nabla \mathcal{L}^0(\theta_t)\| + (\|H_{t-1}^n\| \|u_{t-1}^n\| + \|s_{t-1}^n\|) \left| \frac{(s_{t-1}^n)^\top \nabla \mathcal{L}_n(\theta_t^n)}{(s_{t-1}^n)^\top u_{t-1}^n} - \frac{s_{t-1}^\top \nabla \mathcal{L}^0(\theta_t)}{s_{t-1}^\top u_{t-1}} \right|$$

$$+ (\|H_{t-1}^n\| \|u_{t-1}^n - u_{t-1}\| + \|s_{t-1} - s_{t-1}^n\|) \left| \frac{s_{t-1}^\top \nabla \mathcal{L}^0(\theta_t)}{s_{t-1}^\top u_{t-1}} \right|$$

$$\le C_p \|\theta_{t-1} - \theta^*\|^{2-2p} (C_p + C_p c_t) \|\theta_{t-1} - \theta^*\|^{p-1} \sqrt{d \log(1/\delta)/n}$$

$$+ (C_p \|\theta_{t-1} - \theta^*\|^{2-2p} C_p \|\theta_{t-1} - \theta^*\|^{2p-1} + \|\theta_{t-1} - \theta^*\|)(C_p + C_p c_t + C_p c_{t-1}) \|\theta_{t-1} - \theta^*\|^{-p} \sqrt{d \log(1/\delta)/n}$$

$$+ (C_p \|\theta_{t-1} - \theta^*\|^{2-2p} (C_p + C_p c_t + C_p c_{t-1}) \|\theta_{t-1} - \theta^*\|^{p-1} \sqrt{d \log(1/\delta)/n}$$

$$+ (c_t + c_{t-1}) \|\theta_{t-1} - \theta^*\|^{1-p} \sqrt{d \log(1/\delta)/n}) C_p$$

$$\le (C_p + C_p c_t) \|\theta_{t-1} - \theta^*\|^{1-p} \sqrt{d \log(1/\delta)/n} + (C_p + C_p c_t + C_p c_{t-1}) \|\theta_{t-1} - \theta^*\|^{1-p} \sqrt{d \log(1/\delta)/n}$$

$$+ (C_p + C_p c_t + C_p c_{t-1}) \|\theta_{t-1} - \theta^*\|^{1-p} \sqrt{d \log(1/\delta)/n}$$

$$\le (C_p + C_p c_t + C_p c_{t-1}) \|\theta_{t-1} - \theta^*\|^{1-p} \sqrt{d \log(1/\delta)/n}.$$

Hence, we prove that

$$\|H_t^n \nabla \mathcal{L}_n(\theta_t^n) - H_t \nabla \mathcal{L}^0(\theta_t)\| \le (C_p + C_p c_t + C_p c_{t-1}) \|\theta_{t-1} - \theta^*\|^{1-p} \sqrt{d \log(1/\delta)/n}. \tag{72}$$

Notice that by induction and linear convergence rates (60), we observe that

$$\|\theta_t^n - \theta_t\| \le c_t \|\theta_{t-1} - \theta^*\|^{1-p} \sqrt{d \log(1/\delta)/n} \le c_t \frac{1}{r_{t-1}^{1-p}} \|\theta_t - \theta^*\|^{1-p} \sqrt{d \log(1/\delta)/n}$$

$$\le c_t r_{t-1}^{p-1} \|\theta_t - \theta^*\|^{1-p} \sqrt{d \log(1/\delta)/n} \le c_t \|\theta_t - \theta^*\|^{1-p} \sqrt{d \log(1/\delta)/n}. \tag{73}$$

Therefore, leveraging (65), (72), and (73), we have that

$$\|\theta_{t+1}^n - \theta_{t+1}\| \le \|\theta_t^n - \theta_t\| + \|H_t^n \nabla \mathcal{L}_n(\theta_t^n) - H_t \nabla \mathcal{L}^0(\theta_t)\|$$

$$\le c_t \|\theta_t - \theta^*\|^{1-p} \sqrt{d \log(1/\delta)/n} + (C_p + C_p c_t + C_p c_{t-1}) \|\theta_t - \theta^*\|^{1-p} \sqrt{d \log(1/\delta)/n}$$

$$\le (C_p + C_p c_t + C_p c_{t-1}) \|\theta_t - \theta^*\|^{1-p} \sqrt{d \log(1/\delta)/n}.$$

We define that

$$c_{t+1} = C_p + C_p c_t + C_p c_{t-1}.$$

Then, we have that

$$\|\theta_{t+1}^n - \theta_{t+1}\| \le c_{t+1} \|\theta_t - \theta^*\|^{1-p} \sqrt{d \log(1/\delta)/n}.$$

With the standard recursion, we know that $c_t \le (C_p)^t$ for $C_p$ large enough. Hence, (61) holds for $t + 1$.

### A.5.3 Final conclusion

Therefore, using induction we proved that (61) holds for all $t \geq 1$:

$$\|\theta_t^n - \theta_t\| \leq c_t \|\theta_{t-1} - \theta^*\|^{1-p} \sqrt{d \log(1/\delta)/n},$$

where $c_t = \Theta(C_p^t) = \Theta(\exp(t))$. Notice that

$$\|\theta_t^n - \theta^*\| \leq \|\theta_t^n - \theta_t\| + \|\theta_t - \theta^*\| \leq C_p^t \|\theta_t - \theta^*\|^{1-p} \sqrt{d \log(1/\delta)/n} + \|\theta_t - \theta^*\|.$$

The optimal $T$ with minimum $\|\theta_T^n - \theta^*\|$ should satisfy that

$$C_p^T \|\theta_T - \theta^*\|^{-p} \sqrt{d \log(1/\delta)/n} = C_p,$$

for which we obtain $T = \frac{C \log(n/d \log(1/\delta))}{2(p+1)}$ for some constant $C$ that is independent of $d$ and $n$. Therefore, we prove the final conclusion that

$$\|\theta_T^n - \theta^*\| \leq C'(d \log(1/\delta)/n)^{1/(2p+2)},$$

where $C'$ is a constant that is independent of $d$ and $n$.

## B    Additional experiments for the medium SNR regime

Here we briefly illustrate the behavior of BFGS in Medium SNR regime. We consider the generalized linear model with $d = 50, 100, 500$ and $p = 2$. The inputs are still generated by $\{X_i\}_{i=1}^n$, but $\theta^*$ now is uniformly sampled from the sphere with radius $n^{-1/6}$.

The results are shown in Figure 9. We can see BFGS still converges fast, and the statistical radius of middle SNR regime lies between the High SNR and Low SNR. A rigorous characterization of the statistical radius of middle SNR regime will be left as future work.

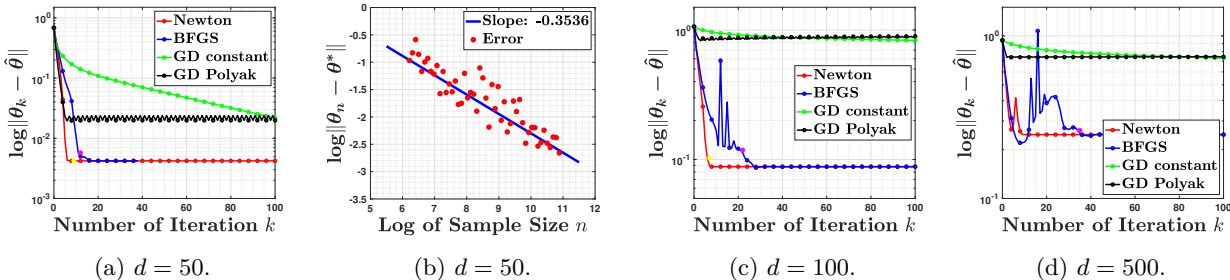

(a) $d = 50$.    (b) $d = 50$.    (c) $d = 100$.    (d) $d = 500$.

Figure 9: Convergence results and statistical results for medium SNR regime with $d = 50$ are shown in (a) and (b). Convergence of different methods with $d = 100$ and $d = 500$ for medium SNR regime are shown in (c) and (d).

