# OpenReview forum: "Statistical and Computational Complexities of BFGS Quasi-Newton Method for Generalized Linear Models"
_TMLR — Accepted by TMLR_

### Review · Reviewer_iVPs · 2023-11-23

**Summary Of Contributions:**

The paper studies the problem of the generalized linear model (GLM), with a polynomial link function. Authors analyze statistical and computational complexities of quasi-Newton (BFGS) method with a main emphasis on the low signal-to-noise ratio (SNR) regime. This work shows theoretical and practical (experimental) benefits of BFGS over Gradient Descent and Newton’s method on synthetic problems.

**Audience:**

Yes

**Broader Impact Concerns:**

No for this work

**Claims And Evidence:**

Yes

**Requested Changes:**

**C1.** I would like to ask the authors to give a precise characterization of the problem being solved in terms of smoothness and strong convexity parameters. BFGS is introduced for the convex setting at the beginning of the paper. It is said that under Assumptions 1 and 2 problem 10 has a unique solution. How far then this problem is from the strongly convex setting? Does it satisfy Polyak-Łojasiewicz (PL) condition?

**C2.** It would be helpful to add more details (maybe from the Appendix) at the beginning of page 5 to make the reasoning more rigorous and clear. It is unclear why the equations are split into multiple lines.

**C3.** The paper would benefit from a more comprehensive overview of the technical innovations and analytical basis of this study, especially in relation to previous works on BFGS. A comparative analysis with prior literature, particularly focusing on the analysis and settings of BFGS, would provide valuable context and highlight the novel contributions of this research. Clarifying these points will help in distinguishing this study from existing works and underscore its originality and significance.

**Strengths And Weaknesses:**

# Strengths
**S1.** Suggested approach (BFGS method) is known for its good practical performance  for practical optimization problems, but until recently has been lacking good theoretical analysis. The idea of using this quasi-Newton method seems very reasonable and worth studying more in the machine learning context. The authors show that their approach improves upon Gradient Descent as its complexity does not depend on the condition number of the problem. Moreover per-iteration cost $\mathcal{O}(d^2)$ is cheaper than Newton’s method $\mathcal{O}(d^3)$.

**S2.** The paper analyses performance of the method on both population (1) and empirical (2) loss with a focus on the low signal-to-noise ratio (SNR) regime. In the first (1) case a linear convergence rate $r_k$ for the iterates of the Algorithm is established. Later $r_k$ is analyzed more accurately and compared with a result for Newton’s method. For the second (2) case of empirical loss, the authors show that minimum distance between iterations and global solution also converges to a statistical radius after a certain number of iterations with high-probability.

**S3.** The authors claim that this is the first global linear convergence analysis of BFGS without line-search for a setting that is neither strictly nor strongly convex. This may be interesting for a broader optimization community as the area of quasi-Newton method have recently attracted novel attention.

**S4.** Overall the paper is well-written and structured, but the clarity and presentation can be improved by expanding and moving some of the explanations from the Appendix to the main part. More on this in the Weaknesses section later.

**S5.** Numerical experiments are well-designed to support the main claims of the work. Several parameter settings are explored both in the main part and in the Appendix.

# Weaknesses

**W1.** Some of the claims made in the paper need to be justified (e.g. by providing a reference where a certain effect was observed)

> the Polyak iterates still have instability during training due to the influence of noise in the models.

> Since we focus on $p \geq 2$ it can be verified that $L(\theta)$ is not strongly convex in a neighborhood of the solution $\theta^*= 0$.

**W1.** **Presentation and clarity**

- Separate notation for the loss $\mathcal{L}$ and function $f$ to be optimized can be confusing.
- Presentation of different settings and special problem cases is somewhat overloaded and lacks clarity
- Second paragraph on page 6 is tricky to understand, especially the statements about error between gradients and Hessians. I believe that some of the Appendix details are needed in the main text.

## Questions
**Q1.** Why does the norm of the optimum $\|\theta^*\|$ has to converge to zero for increasing the sample size $n$.

**Q2.** Why high-probability analysis is needed for Theorem 4?

**Q3.** I would like to ask for a more precise comparison to theoretical results of Newton method.

**Q4.** How exactly the constant step size $\theta$ is tuned “by hand” to achieve the best performance of GD on each problem? Why the “tuned” values are so small: {$10^{-4}, \dots, 10^{-15}$}? Why in the experiments for empirical loss the step size is chosen as $\eta = 0.1$?

**Q5.** What are the main limitations of the proposed approach?

### Minor
At the end of page 4 it is mentioned that $\mathcal{L}(\theta) := \mathbb{E} \mathcal{L}_n (\theta)$ is
> an approximation of that function by its population version

Isn’t it usually the other way around that empirical loss approximates the population one with a finite sample Monte-Carlo estimate?

**Typo** on page 7
> problemequation

---

> ### Author Response · Authors · 2024-03-13
>
> * **Weakness 1.** *Some of the claims made in the paper need to be justified (e.g. by providing a reference where a certain effect was observed). the Polyak iterates still have instability during training due to the influence of noise in the models.*
>
>     **Response.** Done, and thank you for highlighting this issue. We have revised the text to better explain why GD with Polyak step size is not preferable. Here is the updated section in the introduction of the revised paper:
>
>     ``To address this issue, , Ren et al. (2022a) advocated the use of GD with Polyak step size to improve GD's convergence in low SNR scenarios. They demonstrated that the number of iterations becomes a logarithmic function of the sample size. However, since this method remains a first-order approach, its overall complexity is directly proportional to the condition number of the problem. This, in turn, is dependent on both the condition number of the feature vectors' covariance and the norm $\\|\theta^*\\|$. Furthermore, the implementation of the Polyak step size necessitates access to the optimal value of the objective function. As precise estimation of this optimal objective function value may not always be feasible, any inaccuracies could potentially lead to a reduced convergence rate for GD employing Polyak step size."
>
> * **Weakness 2.** *Since we focus on $p \geq 2$, it can be verified that $L(\theta)$ is not strongly convex in a neighborhood of the solution $\theta^\* = 0$.*
>
>     **Response.** Done. Thanks for pointing out this problem. As presented at the top of page 5 in the revised paper, if we calculate the Hessian matrix of the objective function, we notice that the Hessian matrix $\nabla^2{L(\theta)}$ is of the form  $E_X[X(X^\top \theta^*)^{2p - 2}X^\top ]$. Consequently, in a low SNR environment, particularly when the norm of $\theta^\*$ is small, the Hessian matrix becomes highly ill-conditioned and nearly singular. Notably, when  $\theta^* = 0$, the matrix becomes singular, and the strong convexity vanishes. Therefore, $L(\theta)$ is not strongly convex in a neighborhood of the solution $\theta^\* = 0$. For more details, please refer to the blue text added on pages 4 and 5 of the revised paper.
>
> * **Weakness 3.** *Separate notation for the loss $L$ and function $f$ to be optimized can be confusing.*
>
>     **Response.** Done. Thank you for highlighting this issue. We define the objective function $f$ in (10) as a generalized objective function, encompassing the population loss function $\mathcal{L}$ with $\theta^\* = 0$ as a specific instance. This broadens the scope of our findings in Section 4, extending them beyond the limited context of population loss in low SNR scenarios. To alleviate any confusion, we have added an explanatory sentence prior to equation (10). This clarification emphasizes that $f$ represents a more comprehensive version of the population loss function discussed in (9). Consequently, all results presented in Section 4 are applicable to scenarios involving population loss in low SNR regimes.
>
> * **Weakness 4.** *Presentation of different settings and special problem cases is somewhat overloaded and lacks clarity.*
>
>     **Response.** Thank you for highlighting this issue. We acknowledge your point and believe that providing a roadmap or outline can significantly aid readers in navigating the paper. Accordingly, we have incorporated the following outline into the introduction of the revised paper:
>
>     "In Section 2, we discuss the BFGS quasi-Newton method. Section 3 details three scenarios in Generalized Linear Models (GLMs): low, middle, and high SNR regimes, outlining the characteristics of the population loss in each. Section 4 explores BFGS's convergence in low SNR settings, highlighting its linear convergence rate, a marked improvement over gradient descent's sublinear rate. This section also compares the convergence rates of BFGS and Newton's method. Section 5 applies these insights to establish the convergence results of BFGS for the empirical loss $\mathcal{L}_n$ in the low SNR regime. Lastly, our numerical experiments are presented in Section 6."

---

> ### Author Response · Authors · 2024-03-13
>
> * **Weakness 5.** *Second paragraph on page 6 is tricky to understand, especially the statements about error between gradients and Hessians. I believe that some of the Appendix details are needed in the main text.*
>
>     **Response.** Done. Thanks for pointing out this problem. We have modified this paragraph and added a remark to better explain this point. Please check remark 1. The key content of this remark is that analyzing the population loss with $\theta^\* = 0$ and $\\|\theta^{\*}\\| \leq C_{1} (d/n)^{1/(2p)}$ doesn't influence the convergence analysis of the BFGS method applied to the empirical loss. This is because the error between population loss with $\theta^\* = 0$ and $\\|\theta^{\*}\\| \leq C_{1} (d/n)^{1/(2p)}$ is upper bounded by the statistical error between population loss and empirical loss, i.e.,
> $$
>     \qquad \sup_{\theta\in\mathbb{B}(\theta^*, r)}  \\|\nabla \mathcal{L}(\theta) -  \nabla \mathcal{L}^{0}(\theta)\\|  \leq C_p (\\|\theta^\*\\| + r)^{p-1} \sqrt{d\log(1/\delta)/n},
> $$
> $$
>      \qquad \sup_{\theta\in\mathbb{B}(\theta^*, r)} \\|\nabla^2 \mathcal{L}(\theta) -  \nabla^2 \mathcal{L}^{0}(\theta)\\|  \leq C_p (\\|\theta^*\\| + r)^{p-2} \sqrt{d\log(1/\delta)/n},
> $$
> where $\mathcal{L}^{0}$ is the population loss function with $\theta^\* = 0$. We observe that the errors between gradients and Hessians of the population loss with $\theta^\* = 0$ and $\\|\theta^{\*}\\| \leq C_{1} (d/n)^{1/(2p)}$ are upper bounded by the corresponding statistical errors between the population loss and the empirical loss in the low SNR regime, respectively. Therefore, analyzing the population loss with $\theta^\* = 0$ in section 4 instead of analyzing the population loss with $\\|\theta^{\*}\\| \leq C_{1} (d/n)^{1/(2p)}$ doesn't affect the proof of the convergence properties of the BFGS method applied to the empirical loss in section 5. Please check details in the Appendix A.4.
>
> * **Question 1.** *Why does the norm of the optimum $\\|\theta^\*\\|$ has to converge to zero for increasing the sample size.*
>
>     **Response.** The low SNR regime is characterized by the condition $\\|\theta^*\\| \leq C_1(\frac{d}{n})^{2p}$. This definition implies that as the sample size $n$ increases, the norm of the optimal parameter $\theta^*$ approaches zero. Thus, in the low SNR regime, the norm of $\\|\theta^*\\|$ diminishes with an increasing sample size.
>
> * **Question 2.** *Why high-probability analysis is needed for Theorem 4?*
>
>     **Response.** This is a good point. Our analysis hinges on linking the gradient and Hessian of the empirical loss to the population loss, subsequently demonstrating a convergence rate for the empirical loss based on the linear convergence established in Section 4 for the population loss. More precisely, the following uniform concentration inequalities are valid with a probability of $1 - \delta$,
> $$
>     \qquad \sup_{\theta\in\mathbb{B}(\theta^*, r)}  \\|\nabla \mathcal{L}(\theta) -  \nabla \mathcal{L}(\theta)\\|  \leq C_p (\\|\theta^\*\\| + r)^{p-1} \sqrt{d\log(1/\delta)/n},
> $$
> $$
>      \qquad \sup_{\theta\in\mathbb{B}(\theta^*, r)} \\|\nabla^2 \mathcal{L}(\theta) -  \nabla^2 \mathcal{L}(\theta)\\|  \leq C_p (\\|\theta^*\\| + r)^{p-2} \sqrt{d\log(1/\delta)/n},
> $$
>     when $n = \Omega( (d \log d/\delta)^{2p})$. We utilize these high-probability bounds, along with the linear convergence of BFGS in the population case, to establish the convergence guarantee for the empirical loss, as presented in Theorem 4. Further details about these bounds and our convergence analysis are provided in Appendix A.5.
>
> We have highlighted this point in the revised paper. Please refer to the blue text following Theorem 4 in the revised paper. Thank you for your comment.

---

> ### Author Response · Authors · 2024-03-13
>
> * **Question 3.** *I would like to ask for a more precise comparison to theoretical results of Newton method.*
>
>     **Response.**  Done. As outlined in Theorem 3, Section 4, Newton's method achieves global linear convergence to the optimal solution in the low SNR setting when applied to population loss. Notably, the linear convergence contraction factor for Newton's method is provably smaller than that of BFGS, indicating faster convergence. However, the difference in the number of iterations between Newton and BFGS is not significant, as both require $\log(1/\epsilon)$ iterations, where $\epsilon$ denotes the desired accuracy.
>
>     Regarding computational cost, each iteration of Newton's method for minimizing empirical loss with $n$ samples is $\mathcal{O}(nd+nd^2+d^3)$. This cost breaks down into gradient computation (first term), Hessian calculation (second term), and matrix inversion (third term). Meanwhile, BFGS's per-iteration cost is only $\mathcal{O}(nd+d^2)$, involving just gradient computation and a matrix-vector product, significantly reducing its overall computational complexity compared to Newton's method. Further details and experimental comparisons of these methods are available in Section 4.1.
>
> * **Question 4.** *How exactly the constant step size $\eta$ is tuned “by hand” to achieve the best performance of GD on each problem? Why the “tuned” values are so small? Why in the experiments for empirical loss the step size is chosen as $\eta = 0.1$?*
>
>     **Response.** Done. Thank you for highlighting this issue. To fine-tune the parameter $\eta$, we employed a manual grid search across the following values $[10^{-15}, 10^{-9}, ..., 10^{-4}]$. We tested different step sizes for $\eta$ and selected the value that yielded the best performance in Gradient Descent (GD) for each specific problem. In the experiments involving empirical loss, we found that $\eta = 0.1$ delivered the most effective results with the empirical loss function. This point is mentioned in the revised paper.
>
> * **Question 5.** *What are the main limitations of the proposed approach?*
>
>     **Response.** This is a great question. One notable drawback of using the BFGS method for solving generalized linear model problems is its computational cost per iteration, which is $\mathcal{O}(d^2)$. This is indeed higher than the $\mathcal{O}(d)$ cost associated with first-order methods like Gradient Descent (GD). However, BFGS compensates for this higher cost with a faster convergence rate. It requires only a logarithmic number of iterations to reach the statistical radius, whereas GD needs a polynomial number of iterations to achieve the same level of error.
>
> * **Question 6.** *At the end of page 4 it is mentioned that $L(\theta) = E[L_n(\theta)]$ is an approximation of that function by its population version. Isn’t it usually the other way around that empirical loss approximates the population one with a finite sample Monte-Carlo estimate?*
>
>     **Response.** Done. Thanks for pointing out this issue. We agree with the reviewer's point that it is indeed the empirical loss which approximates the population loss using a finite sample Monte-Carlo estimate. We have updated the text to reflect this accurate representation in the revised version of the paper. Please review the corrected paragraph between equations (7) and (8).
>
> * **Question 7.** *Typo on page 7 problemequation*
>
>     **Response.** Done. Thanks for raising this point. We have fixed this typo.

---

> ### Author Response · Authors · 2024-03-13
>
> * **Requested Changes 1.** *I would like to ask the authors to give a precise characterization of the problem being solved in terms of smoothness and strong convexity parameters. BFGS is introduced for the convex setting at the beginning of the paper. It is said that under Assumptions 1 and 2 problem 10 has a unique solution. How far then this problem is from the strongly convex setting? Does it satisfy Polyak-Łojasiewicz (PL) condition?*
>
>     **Response.** We appreciate the reviewer raising this question. For problem (10), the Hessian of the objective function is outlined in (25). It is evident that the Hessian $\nabla^2{f(\theta)}$ is always semi-positive definite. It becomes singular if and only if $A\theta = b$. Thus, the problem remains strongly convex, satisfying the PL condition as long as $\theta$ does not meet the condition $A\theta = b$. Conversely, if $A\theta = b$ is satisfied, the Hessian matrix becomes zero, indicating the absence of strong convexity and the PL condition. Additionally, when $\\|A\theta - b\\|$ is very small, the problem becomes highly ill-conditioned due to the near singularity of the Hessian matrix. Consequently, the strong convexity and the PL condition of the objective function in (10) diminish as the error $\\|A\theta - b\\|$ approaches zero. In summary, for the optimization problem in (10), when $\\|A\theta - b\\| > 0$, the objective function is strongly convex and meets the PL condition. However, when $\\|A\theta - b\\| = 0$, the problem's Hessian is singular, and both strong convexity and the PL condition are lost. This is indeed the case when we focus on the low SNR setting.
>
> * **Requested Changes 2.** *It would be helpful to add more details (maybe from the Appendix) at the beginning of page 5 to make the reasoning more rigorous and clear. It is unclear why the equations are split into multiple lines.*
>
>     **Response.** Done. Thanks for pointing out this issue. We have incorporated additional details from the Appendix into the main text for a clearer illustration of these equations. To enhance readability, we arranged all the equations on a single line. Additionally, we included content regarding the Taylor expansion of the objective function $L(\theta)$ in the high SNR regime. For further details, please refer to the paragraphs and equations at the beginning of page 5 in the revised version of the paper.

---

> ### Author Response · Authors · 2024-03-13
>
> * **Requested Changes 3.** *The paper would benefit from a more comprehensive overview of the technical innovations and analytical basis of this study, especially in relation to previous works on BFGS. A comparative analysis with prior literature, particularly focusing on the analysis and settings of BFGS, would provide valuable context and highlight the novel contributions of this research. Clarifying these points will help in distinguishing this study from existing works and underscore its originality and significance.*
>
>     **Response.** Done. In the final paragraph of Section 2, we have detailed the historical development of the BFGS method, highlighting recent progress in non-asymptotic superlinear analysis of quasi-Newton methods. As mentioned in the paper, it is crucial to note that these superlinear convergence analyses presuppose \textbf{strong convexity} in the objective function, a condition \textbf{not met in the low SNR setting} of generalized linear models, as explained in our response to the previously mentioned weakness 2. The potential singularity of the Hessian at the optimal solution leads to the absence of strong convexity in the low SNR regime. This necessitates the development of novel analyses for applying the quasi-Newton method to generalized linear models in this specific context. We have emphasized this point in the blue-highlighted text of the last paragraph in Section 2.
>
>     Additionally, in the revised paper, after Theorem 1, we elaborate on how we utilize the structure of the considered convex problem (which does not fulfill the strong convexity criterion) to establish the global convergence of BFGS for minimizing the population loss in the low SNR setting. Specifically, we have added the following text:
>
>     "We should highlight the above linear convergence  result and our convergence analysis both rely heavily on the distinct structure of problem (10) and may not hold for a general convex minimization problem. Specifically, it can be shown that if we had access to the exact Hessian and could perform a Newton's update to solve the problem in problem (10), then the error vectors *$(\theta_k - \hat{\theta})_{k \geq 0}$* would all be parallel. This property arises due to the fact that for the problem in problem (10)  the Newton direction is $\nabla ^2f(\theta_{k-1})^{-1}\nabla f(\theta_{k-1})= (\theta_{k-1}- \hat{\theta}) - \frac{q-2}{q-1} (\theta_{k-1}- \hat{\theta})$. Consequently, the next error vector $\theta_{k-1}- \hat{\theta}$ computed by running one step of Newton would satisfy $\theta_{k}- \hat{\theta}= \frac{q-2}{q-1} (\theta_{k-1}- \hat{\theta})$. Therefore, the error vector at time $k$, denoted by $\theta_{k}- \hat{\theta}$, is parallel to the previous error vector $\theta_{k-1}- \hat{\theta}$, with the only difference being that its norm is reduced by a factor of $\frac{q-2}{q-1}$. Using an induction argument, it simply follows that all error vectors $(\theta_k - \hat{\theta})_{k \geq 0}$ remain parallel to each other while their norm contracts at a rate of $\frac{q-2}{q-1}$. This is a key point that is used in the result for Newton's method in Theorem~3."
>
>     In the convergence analysis of BFGS (stated in the proof of Theorem~1), we use a similar argument. While we cannot guarantee that the Hessian approximation matrix in the BFGS method matches the exact Hessian, we demonstrate that it can inherit this property, maintaining all error vectors *$(\theta_k - \hat{\theta})_{k \geq 0}$* as parallel to each other. Additionally, we show that the norm is shrinking at a factor of $r_k<1$, which is always larger than $\frac{q-2}{q-1}$, yet remains independent of the problem's condition number or dimensions. In the following theorem, we show that for $q \geq 4$, the linear rate contraction factors *$\\{r_k\\}_{k = 0}^{\infty}$* also converge linearly to a fixed point contraction factor $r_*$ determined by the parameter $q$."

---

### Review · Reviewer_nEq8 · 2024-01-03

**Summary Of Contributions:**

This paper studies the time complexity of the BFGS algorithm in a specific generalized linear model. It shows that when the link function of the GLM is $x^p$, the algorithm converges to a radius of $n^{1/(2p)}$ around the true signal vector in $O(\log(n))$ iterations, where $n$ is the number of samples.

The proof proceeds in two steps: first, it studies the BFGS algorithm for infinite sample complexity and zero signal, and shows a linear convergence rate. Then, for large enough sample complexity, it bounds the difference between the Hessian of the finite and infinite sample setting, and leverages those bounds to show convergence of the actual BFGS algorithm.

The authors also provide numerical experiments showing that the BFGS algorithm outperforms classical Gradient Descent, even with optimal step-sizes.

**Audience:**

Yes

**Claims And Evidence:**

Yes

**Requested Changes:**

All mentioned changes are mostly for improvement.

- you should use `\eqref`for referring to equations, and check some of the references (e.g. "empirical loss equation 7" would be better as "empirical loss (7)")
- you mention extensively a notion of "statistical radius", that is never defined in the paper; it does not seem to be an actual lower bound for recovering $\theta^\star$, so what does it correspond to ?
- do you have any lower bound for the complexity of learning a GLM with link function $x^p$, if we want to be independent from the condition number of $\Sigma$ ?
- top of page 5 : your obtained expression for $\mathcal L(\theta)$ seems convex independently from the norm of $\theta^\star$ (and its relation to $\sigma$), so the explanation is a bit unconvincing.

**Strengths And Weaknesses:**

This paper provides a nice analysis of the BFGS algorithm, which is usually hard to study, especially without line search. Although the proofs are quite computation-heavy, they are presented nicely enough to be understood and checked without too much effort. Thus, this article completely satistfies the criteria for a TMLR publication.

My main issue with the paper (outside of minor qualms in the presentation) is the bound on sample complexity that you obtain, which is in $n \asymp (d\log(d))^{2p}$. This seems extremely wasteful, since we can learn a polynomial of degree $p$ with many methods in $O(d^p)$ samples, and (since $x^p$ has information exponent 1) [this paper](https://www.jmlr.org/papers/volume22/20-1288/20-1288.pdf) implies that if we know the link function, $O(d)$ samples are enough. Further, since we need to compute gradients of $\mathcal L_n$, I feel like the actual complexity of one BFGS iteration is $O(n \cdot d^2)$, which makes the previous issue all the more salient.

---

> ### Author Response · Authors · 2024-03-13
>
> * **Weakness 1.** *My main issue with the paper (outside of minor qualms in the presentation) is the bound on sample complexity that you obtain, which is in $n = (d\log{d})^{2p}$. This seems extremely wasteful, since we can learn a polynomial of degree $p$ with many methods in $\mathcal{O}(d^p)$ samples, and (since $x^p$ has information exponent 1) this paper implies that if we know the link function, $\mathcal{O}(d)$ samples are enough. Further, since we need to compute gradients of $L_n$, I feel like the actual complexity of one BFGS iteration is $\mathcal{O}(nd^2)$, which makes the previous issue all the more salient.*
>
>     **Response.** These are excellent points, and we will try to address them in the following sentences. First, we would like to clarify that the per iteration complexity of BFGS is $\mathcal{O}(nd + d^2)$ instead of $\mathcal{O}(nd^2)$. Note that the update of BFGS requires gradient calculation that has a cost of the order $\mathcal{O}(nd)$ and a matrix-vector product that has a cost of $\mathcal{O}(d^2)$. Hence, the overall cost per iteration is $\mathcal{O}(nd + d^2)$. Note that this cost is comparable with the $\mathcal{O}(nd)$ cost of GD when we are in the regime where $n\geq d$.
>
>     Regarding the sample complexity requirement stated in Theorem 4, which necessitates $n = (d\log{d})^{2p}$, we acknowledge the reviewer's suggestion that it might be feasible to establish a similar upper bound with a more favorable dependency on the polynomial link function $p$. However, it is important to emphasize that the primary aim of our paper isn't to achieve optimal sample complexity. Our main objective is to demonstrate that, with an appropriate number of samples, **BFGS can solve the problem more efficiently than GD and GD with Polyak step size.** In fact, the conditions we set for the number of samples and convergence radius are comparable to those in [Ren et al. (2022a)], which analyzes the convergence of GD with Polyak step size. Nonetheless, we agree with the reviewer that exploring the potential to improve the conditions on the number of required samples presents an intriguing statistical challenge warranting further investigation.
>
> * **Requested Changes 1.** *you should use 'eqref' referring to equations, and check some of the references (e.g. "empirical loss equation 7" would be better as "empirical loss (7)")*
>
>     **Response.** Done. Thank you for pointing out this issue. We have revised the paper, using 'eqref' to refer to all the equations. Please review the equation references in the updated submission.
>
> * **Requested Changes 2.** *you mention extensively a notion of "statistical radius", that is never defined in the paper; it does not seem to be an actual lower bound for recovering $\theta^\*$, so what does it correspond to ?*
>
>     **Response.** Done. Your comment is indeed valid. As you correctly pointed out, there is no established lower bound for this setting, which could represent the ultimate statistical accuracy. A more accurate approach would be to refer to the upper bound on the right-hand side of Theorem 4, which pertains to the norm of the difference between the parameter estimates of the empirical loss and the true parameters. We have revised our paper to address this issue. Thank you for your comment.

---

> ### Author Response · Authors · 2024-03-13
>
> * **Requested Changes 3.** *do you have any lower bound for the complexity of learning a GLM with link function $x^p$, if we want to be independent from the condition number of $\Sigma$?*
>
>     **Response.**  This is an excellent point. We are not aware of a lower bound for the sample complexity of learning a GLM with a polynomial link function. That said, it is important to emphasize that the primary objective of this paper is not to determine the optimal sample complexity for learning a GLM with a polynomial link function. Rather, our aim is to identify an optimization method for achieving a predetermined level of accuracy more efficiently than both Gradient Descent (GD) and GD enhanced with Polyak's step sizes, as explored in [Ren et al. (2022a)].
>
> * **Requested Changes 4.** *top of page 5 : your obtained expression for $L(\theta)$ seems convex independently from the norm of $\theta^\*$ (and its relation to $\sigma$), so the explanation is a bit unconvincing.*
>
>     **Response.** Done. Thank you for highlighting this issue. Upon calculating the Hessian matrix of the objective function, as defined at the top of page 5, we notice that the Hessian matrix $\nabla^2{L(\theta)}$ is of the form  $E_X[X(X^\top \theta^*)^{2p - 2}X^\top ]$. Consequently, in cases where $\\|\theta^*\\| \geq C\sigma$, the smallest eigenvalue of the Hessian  $\nabla^2{L(\theta)}$ is significantly greater than zero, confirming that the matrix is strictly symmetric positive definite.   This indicates that the objective function exhibits strong convexity in scenarios with a high signal-to-noise ratio (SNR).  Conversely, in a low SNR environment, particularly when the norm of $\theta^*$ is small, the Hessian matrix becomes highly ill-conditioned and nearly singular. Notably, when  $\theta^* = 0$, the matrix becomes singular, and the strong convexity vanishes. We have elucidated this relationship between the convexity of $L(\theta)$ with the norm of $\theta^*$ (and its relation to $\sigma$) by incorporating additional equations at the beginning of page 5 in the revised submission.

---

### Review · Reviewer_uJae · 2024-01-11

**Summary Of Contributions:**

The work studies statistical and computational complexities of the classical BFGS method applied to generalized linear models (GLM). In particular, the authors focus on low signal-to-noise ratio (SNR) regime, where the loss function is merely convex (not strictly convex). In this regime, the loss reduces to least squares problem with arbitrary power $q \geq 4$, i.e., $f(\theta) = \\| A \theta - b\\|^p$, where $A^{\top} A$ is invertible. Utilizing this special structure, the authors prove global linear convergence of BFGS method. Importantly, the linear rate is independent of the condition number, although only first-order information is utilized. The work also shows (faster) global convergence of Newton method for this problem and extends the result for BFGS to finite sample setting.

**Audience:**

Yes

**Broader Impact Concerns:**

---

**Claims And Evidence:**

Yes

**Requested Changes:**

Addressing points 2., 3., 4. and 7. in the weaknesses above would be the most critical for acceptance.

**Strengths And Weaknesses:**

**Strengths:**

1. The authors approach a theoretically important problem of understanding convergence of BFGS algorithm on a GLM problem without strict convexity. The proof of the main results (Theorems 1 and 2) seems solid and easy to follow.

2. The paper introduces the problem well explaining the background and the technical challenges.


**Weaknesses:**

1. The statement in the second sentence of the abstract seems contradictory. It says GD is optimal in high SNR regime, but it is later revealed that in the same regime BFGS is known to be faster, achieving super-linear convergence due to strong convexity. Of course, the iteration cost are different, but it should be clarified what is meant by optimality here.

2. Although vanilla GD might converge sub-linearly on problem (10), it is known that mirror descent with a distance generating function $h(x) = \frac{1}{q + 2} \\|x\\|^{q+2} +  \frac{1}{ 2} \\|x\\|^{2}$ will converge linearly, see e.g., Lu et al. (2018). It would make sense to compare to such approach both in theory and experiments. Notice that the iteration complexity is merely $O(d)$ for mirror descent.

3. The property of BFGS and the Newton method that the iterates $(\theta_k - \hat \theta )_{k\geq0}$ are always parallel to each other seems surprising to me (see the last paragraph of Appendix A.1). Is it a common property for quasi-newton methods? If not, it would make sense to elaborate more on this property in the main part rather than merely commenting in the proof. Some illustration in small experiment would help dispel the doubts.

4. The proof of Theorem 2 in appendix A.2 is not complete. For $q \leq 11$, only integer values are checked by the picture (Fig. 3). However, the property might fail for non-integer $q$.

5. In Equations (70), (71), (72), (73) the inverse of the gradient is taken. I assume it's supposed to be a Hessian. In equations (104) and (105), the absolute value is missing for the last terms (are they positive?).

6. Markers should be added to all pictures.

7. I was not able to parse the proof in section A.4. Some revision of it would be helpful. The main problem is that it is 7 pages without any structure. E.g., splitting the proof into lemmas could be very helpful.

8. On page 19, referring to a specific lemma in Ren et al. (2022b) would help.

**Questions:**

1. Is it possible to show global linear convergence of BFGS for a general convex function without the special structure (10)?

2. Authors explain that middle SNR regime is technically more challenging, but is there any negative results for this setting? E.g., does the problem have spurious local minima or saddle points? I think it would make sense to draw connections with the literature on linear neural networks, e.g., Kawaguchi (2016).


**Typos and other suggestions:**

1. Page 26 "Hign". Page 18 "inequation". Page 19 "simply the proof"

2. Consider removing the equation counter where it is not necessary.

Haihao Lu, Robert M Freund, and Yurii Nesterov. Relatively smooth convex optimization by first- order methods, and applications. SIAM Journal on Optimization, 28(1):333–354, 2018.

Kenji Kawaguchi. Deep learning without poor local minima. Advances in neural information processing systems, 29, 2016.

---

> ### Author Response · Authors · 2024-03-13
>
> * **Weakness 1.**  *The statement in the second sentence of the abstract seems contradictory. It says GD is optimal in high SNR regime, but it is later revealed that in the same regime BFGS is known to be faster, achieving super-linear convergence due to strong convexity. Of course, the iteration cost are different, but it should be clarified what is meant by optimality here.*
>
>   **Response.** Done. This is a valid point, and in light of your suggestion, we have revised the abstract. Please review the updated abstract. Thank you for your comment.
>
> * **Weakness 2.** *Although vanilla GD might converge sub-linearly on problem (10), it is known that mirror descent with a distance generating function $h(x) = \frac{1}{q + 2}\\|x\\|^{q + 2} + \frac{1}{2}\\|x\\|^2$ will converge linearly, see e.g., Lu et al. (2018). It would make sense to compare to such approach both in theory and experiments. Notice that the iteration complexity is merely $\mathcal{O}(d)$ for mirror descent.*
>
>     **Response.** Done. Thank you for referring to this paper. As you correctly pointed out, Lu et al. (2018) demonstrated that Mirror Descent, with the appropriate choice of Bregman Distance, is able to solve the problem in (10) at a linear rate, thus improving the convergence rate of GD. However, since it lacks any form of curvature approximation, its convergence rate still depends on the condition number $\kappa$ of the problem and takes the form $1 - \frac{1}{\kappa}$.
>
>     Contrary to this result, the convergence results presented in our paper in Theorems 1 and 2 show that running BFGS on problem (10) achieves a linear convergence rate, which is independent of both the condition number $\kappa$ and the problem dimension $d$, highlighting the advantage of BFGS over first-order methods such as Mirror Descent.
>
>     Considering these points, in scenarios like the low SNR for generalized linear models where the problem is ill-conditioned, the linear convergence rate of the Mirror Descent method could be significantly slower compared to that of the BFGS algorithm. We have added Remark 3 in the revised paper and included the aforementioned discussion about the comparison between Mirror Descent and BFGS. Also, in the introduction of the revised paper, we added a paragraph about the possibility of using mirror descent to achieve a linear rate for the considered problem.
>
>     Furthermore, following your suggestion, we have incorporated Mirror Descent in our numerical experiments. As expected, it does converge at a linear rate in the low SNR setting, but the rate of convergence is slower than that of BFGS. Please check the updated plots in Figure 1.
>
>     Thank you for raising this point.

---

> ### Author Response · Authors · 2024-03-13
>
> * **Weakness 3.** *The property of BFGS and the Newton method that the iterates $(\theta_k - \hat{\theta})_{k \geq 0}$ are always parallel to each other seems surprising to me (see the last paragraph of Appendix A.1). Is it a common property for quasi-newton methods? If not, it would make sense to elaborate more on this property in the main part rather than merely commenting in the proof. Some illustration in small experiment would help dispel the doubts.*
>
>     **Response.** Done. This is an excellent point. As the reviewer has noted, in our convergence analysis we demonstrate that the error vectors  *$(\theta_k - \hat{\theta})_{k \geq 0}$* are always parallel to each other for both Newton's method and BFGS. This is indeed true due to the specific structure of the considered problem and does not hold for general convex or strongly convex problems. This property arises due to the fact that the Newton direction $\nabla ^2f(\theta_{k-1})^{-1}\nabla f(\theta_{k-1})$ can be expressed as
>     $$
>     \nabla ^2f(\theta_{k-1})^{-1}\nabla f(\theta_{k-1})= (\theta_{k-1}- \hat{\theta}) - \frac{q-2}{q-1} (\theta_{k-1}- \hat{\theta})
>     $$
>     Consequently, the next error vector $\theta_{k-1}- \hat{\theta}$ computed by running one step of Newton would satisfy
>     $$
>     \theta_{k}- \hat{\theta}= \theta_{k-1}-  \nabla ^2f(\theta_{k-1})^{-1}\nabla f(\theta_{k-1})- \hat{\theta}= \frac{q-2}{q-1} (\theta_{k-1}- \hat{\theta})
>     $$
>     Therefore, the error vector at time $k$, denoted by $\theta_{k}- \hat{\theta}$, is parallel to the previous error vector $\theta_{k}- \hat{\theta}$, with the only difference being that its norm is reduced by a factor of $\frac{q-2}{q-1}$. Using an induction argument, it simply follows that all error vectors $(\theta_k - \hat{\theta})_{k \geq 0}$ remain parallel to each other while their norm contracts at a rate of $\frac{q-2}{q-1}$.
>
>     In the analysis of BFGS, while we cannot guarantee that the Hessian approximation matrix matches the exact Hessian, we show that it can inherit this property and keep all the error vectors $(\theta_k - \hat{\theta})_{k \geq 0}$ parallel to each other. The norm is shrinking with a factor of $r_k<1$ that is always larger than $\frac{q-2}{q-1}$, but is independent of the problem's condition number or dimensions.
>
>     As the reviewer pointed out, the specific property of the objective function plays a significant role in our convergence analysis, and we have highlighted this point in the blue text after Theorem 1 in the revised paper. Moreover, to better visualize this property, we have plotted $\cos{\theta_k} = \frac{(\theta_k - \hat{\theta})^\top(\theta_0 - \hat{\theta})}{\\|\theta_k - \hat{\theta}\\|\\|\theta_0 - \hat{\theta}\\|}$ for $k \geq 1$ for both Newton's method and BFGS in Appendix A.1. As expected, the value of the cosine for both methods is always~$1$, indicating that the error vectors $(\theta_k - \hat{\theta})_{k \geq 0}$ are always parallel to each other.
>
>     Please check the added blue text after Theorem 1 and before Theorem 2 in the revised paper for more details. Thanks for raising this point.
>
> * **Weakness 4.** *The proof of Theorem 2 in appendix A.2 is not complete. For $q \leq 11$, only integer values are checked by the picture (Fig. 3). However, the property might fail for non-integer $q$.*
>
>     **Response.** Done. Thank you for highlighting this issue. We agree with the reviewer that for $q \leq 11$, the property may not hold for non-integer values of $q$. However, our paper primarily focuses on positive integer values of $q$. This focus is due to the generalized linear model with the polynomial link function, as defined in Equation (6), which necessitates that $p$ be an integer. Given the relationship $q = 2p \geq 4$, as outlined in problem (10), it follows that the value of $q$ is also an integer. We highlighted this point in the revised paper. Please check the blue text after equation (10).
>      Also, in the proof of Theorem~2, presented in Appendix A.2, we have meticulously verified all integer values of $q$ within the range of $4 \leq q \leq 11$, as illustrated in Figure 3. Additionally, we have proven all the results for any real number value of $q \geq 12$ which indeed implies it holds for any integer $q \geq 12$. To clarify, we have included a condition in the statement of Theorem 2 in the revised submission, specifying that $q$ is an integer.
>
> * **Weakness 5.** *In Equations (70), (71), (72), (73) the inverse of the gradient is taken. I assume it's supposed to be a Hessian. In equations (104) and (105), the absolute value is missing for the last terms (are they positive?).*
>
>     **Response.** Done. Thank you for catching these typos. In Equations (70) to (73), the term should be the inverse of the Hessian, not the gradient. We have corrected this error. Additionally, in Equations (104) and (105), absolute values are required for the last terms, which we have now added. We appreciate your attention to these details.

---

> ### Author Response · Authors · 2024-03-13
>
> * **Weakness 6.** *Markers should be added to all pictures.*
>
>      **Response.** Done. Thank you for the advice. We have now added markers to all the pictures.
>
> * **Weakness 7.** *I was not able to parse the proof in section A.4. Some revision of it would be helpful. The main problem is that it is 7 pages without any structure. E.g., splitting the proof into lemmas could be very helpful.*
>
>      **Response.** Done. We have thoroughly revised the structure of the proof in Section A.4. This revision involved dividing the proof into shorter sections, thereby making it clearer and more comprehensible. Previously, the excessive length of the proof might have posed challenges for reviewers in understanding it. We have also made explicit that the result is derived using an induction argument. To facilitate a clearer presentation of this argument, we introduced multiple intermediate lemmas. This approach should make the argument easier to follow. Please review the revised proof. Thank you for your feedback.
>
> * **Weakness 8.** *On page 19, referring to a specific lemma in Ren et al. (2022b) would help.*
>
>      **Response.** Done. Thanks for pointing out this issue. We have modified this specific reference in the revised version of the paper. We referred to the Lemma 6 in Ren et al. (2022b).
>
> * **Question 1.** *Is it possible to show global linear convergence of BFGS for a general convex function without the special structure (10)?*
>
>      **Response.** This is an excellent question. Unfortunately, the global convergence analysis of BFGS for general strongly convex functions remains an unresolved issue, despite being a topic of discussion in the field. The existing studies on this matter, particularly in the context of general strongly convex and smooth settings, only offer non-asymptotic guarantees within a local neighborhood of the solution. For further details, please refer to the paper by Jin, Qiujiang, and Aryan Mokhtari, titled "Non-asymptotic superlinear convergence of standard quasi-Newton methods," published in Mathematical Programming (200.1, 2023, pp. 425-473), and the references cited therein.
>
>     Moreover, in the specific scenario we considered, where the objective function is not even strongly convex, there is no known global complexity that is better than gradient descent for general loss functions. In this submission, we rely heavily on the unique structure of the Generalized Linear Model (GLM) to establish our global linear convergence results for a non-strongly convex function, as highlighted in the response to one of your previous questions.
>
>     A global linear convergence analysis of BFGS, applicable to general convex, strictly convex, or even strongly convex functions, and that provably outperforms gradient descent-type methods, remains an open question. This intriguing area warrants future research and certainly requires further exploration.
>
> * **Question 2.** *Authors explain that middle SNR regime is technically more challenging, but is there any negative results for this setting? E.g., does the problem have spurious local minima or saddle points? I think it would make sense to draw connections with the literature on linear neural networks, e.g., Kawaguchi (2016).*
>
>      **Response.** This is a valid point. Essentially, for our analysis to be effective, it is necessary to remain within the lower SNR  regime. This requirement enables us to utilize the function approximation method described in the paper. Transitioning to the medium SNR regime would violate a condition akin to strict convexity, rendering our approach inapplicable.
>
>     However, it is noteworthy that in our numerical experiments conducted in the medium SNR regime, as presented in Appendix B, we encountered no issues with bad local minima or saddle points. In these cases, the BFGS algorithm consistently converged to the ground truth solution. This suggests that a more thorough examination of BFGS in this context, potentially employing alternative techniques, might pave the way for extending our results, or ones similar, to the medium SNR setting.
>
>     Additionally, it would be insightful to explore the relationship between this setting and existing literature on linear neural networks. Such an investigation could form an interesting direction for future research. Thank you for your insightful comment.
>
> * **Typos and other suggestions 1.** *Page 26 "Hign". Page 18 "inequation". Page 19 "simply the proof"*
>
>      **Response.** Done. Thanks for pointing out this problem. We have fixed these typos in the modified submission.
>
> * **Typos and other suggestions 2.** *Consider removing the equation counter where it is not necessary.*
>
>      **Response.** Done. In the revised version of the paper, we have eliminated the equation counter where it is not necessary. Thank you for highlighting this matter.

---

### Author Response · Authors · 2024-01-26
**Delay of Rebuttal to the Reviews**

Dear editors and reviewers,

Thanks for providing these constructive feedbacks. However, due to the approaching deadline of ICML 2024 and COLT 2024, we are unable to respond to these reviews within two weeks. We ask for a two week delay regarding our rebuttals. We apologize for any inconvenience caused by us. Thank you very much.

---

### Author Response · Authors · 2024-04-15
**Submission of Camera Ready Version**

Dear editor,

Thank you very much for providing all the constructive feedbacks through the entire review process and the final decision. We appreciate all your time and effort for all valuable reviews and advice.

We have submitted the camera ready version of our paper as the pdf file. If you need any other information or action from us, please let us know.

Thanks again for the acceptance of our submission.

Best,
Authors

---

### Decision · Action_Editor_uLk3 · 2024-04-04

**Recommendation:** Accept as is

**Comment:**

This paper investigates the convergence of the BFGS method in generalized linear problems. Unlike previous studies, this paper does not depend on the assumption of strong convexity. Instead, it leverages that the BFGS iterates remain within the linear span of the data vectors if the method is initialized in a specific way. This approach enables the derivation of novel convergence results, particularly in low signal-to-noise (SNR) settings.

This contribution makes an essential step towards a better understanding of quasi-Newton methods.

**Audience:**

This work meets the acceptance criteria of the TMLR: the results are of interest to theoretical optimization researchers, as the theoretical analysis of BFGS is an important challenge. The results are also of interest to a broader optimization community, as the field of quasi-Newton methods has recently attracted new attention.

**Claims And Evidence:**

This paper provides a theoretical contribution to the convergence analysis of quasi-Newton methods for generalized linear models. The assumptions are clearly stated, and mathematical proofs support the claims.

While the reviewers had some initial concerns, the authors responded fully to the feedback and clarified their claims.